# Seismic evidence for a mantle suture and implications for the origin of the Canadian Cordillera

Yunfeng Chen [1,6], Yu Jeffrey Gu[1], Claire A. Currie [1], Stephen T. Johnston [2], Shu-Huei Hung [3], Andrew J. Schaeffer [4,5] & Pascal Audet [4]

The origin of the North American Cordillera and its affinity with the bounding craton are subjects of contentious debate. The mechanisms of orogenesis are rooted in two competing hypotheses known as the accretionary and collisional models. The former model attributes the Cordillera to an archetypal accretionary orogen comprising a collage of exotic terranes. The latter, less popular view argues that the Cordillera is a collisional product between an allochthonous ribbon microcontinent and cratonic North America. Here we present new seismic evidence of a sharp and structurally complex Cordillera–craton boundary in the uppermost mantle beneath the southern Canadian Cordillera, which can be interpreted as either a reshaped craton margin or a Late Cretaceous collisional boundary based on the respective hypotheses. This boundary dips steeply westward underneath a proposed (cryptic) suture in the foreland, consient with the predicted location and geometry of the mantle suture, thus favoring a collisional origin.

[1] Department of Physics, University of Alberta, Edmonton, AB T6G 2E1, Canada. [2] Department of Earth and Atmospheric Sciences, University of Alberta, Edmonton, AB T6G 2E3, Canada. [3] Department of Geosciences, National Taiwan University, Taipei 10617, Taiwan. [4] Department of Earth and Environmental Sciences, University of Ottawa, Ottawa, ON K1N 6N5, Canada. [5] Geological Survey of Canada, Pacific Division, University of Ottawa Sidney, British Columbia, Canada. [6]Present address: Deep Earth Imaging, Future Science Platform, CSIRO, Perth, Australia. Correspondence and requests for materials should be addressed to Y.C. (email: yunfeng1@ualberta.ca)

The Precambrian Laurentia craton, the core of the North American continent, is flanked to the west by the North American Cordillera[1], a broad Phanerozoic orogenic belt that extends from Mexico northwards to Alaska. The Canadian portion of the Cordilleran orogeny was initiated by earliest Cambrian (~540 Ma) rifting and passive margin formation[2], followed by the development of a convergent margin and subsequent Mesozoic and Cenozoic collisional events[3–6]. Various models (e.g., retro-arc thrusting[7], flat slab subduction[8], archipelago convergence[9] and ribbon continent collision[4]) have been proposed to explain this protracted (150–50 Ma) orogenic period, with arguments centering on the provenance, extent and geometry of the accreted terranes that make up the Cordillera[9–13]. The prevailing idea supports the successive emplacements of thin crustal flakes (exotic terranes[3]) over the autochthonous craton margin since at least the Early Jurassic[13–15]. In this scenario, the ancient Laurentian craton constitutes the upper plate above an east-directed subducting Farallon slab. Alternative hypotheses favor episodes of westward subduction of oceanic plates that produced the Cordilleran composite (upper plate) in the form of intra-oceanic arcs (i.e., Insular terrane)[16] or a preassembled micro-continent[4,11,17] prior to collision with the craton. In short, the presumed roles of the Cordilleran terranes and their affiliation to the bounding craton largely decide the styles of the orogenesis (accretionary versus collisional) during the Mesozoic growth of the North American continent.

Keys to differentiating these models are the nature, location and geometry of the boundary between the Cordillera and craton[4,18]. The accretionary model, with its subsurface structures mainly constrained by deep crustal seismic reflection/refraction surveys[15], suggests that much of the Cordillera is built upon a continuous cratonic basement of North America bounded beneath by a landward (eastward) dipping mantle lithosphere[15] and extends as far west as the Coast Belt[13] (Fig. 1). This unique boundary geometry could reflect a destructive (i.e., reshaped) margin that was initially formed by rifting[13] and later modified by episodic lithospheric removal events[22,23]. This model and its inferred boundary processes have ostensibly become a textbook example of an accretionary orogenic belt. In contrast, the collisional model differs from the accretionary concept by predicting both a late (Cretaceous) terminal collision along a (cryptic) suture in the orogenic foreland[4] and a lithospheric scale boundary between the Cordillera and North America[4,11]. The boundary potentially preserves an oceanward (i.e., westward) dipping geometry of a relic craton margin following the break-off of a westward subducting oceanic plate[4,17] (Fig. 1b). Based on a range of geological and geophysical (primarily paleomagnetic) observations[4], the suture (boundary) is assumed to run along, or adjacent to, a carbonate-shale (C-S) facies boundary directly east of the Rocky Mountain Trench (RMT), an orogen-parallel valley that extends from Montana to Yukon with its southern segment formed primarily through Cenozoic normal-faulting (Fig. 1). Although both models provide a tectonic framework for the Canadian Cordillera, they differ in terms of the predicted subsurface structures and processes. Consequently, a better knowledge of the Cordillera–craton boundary (CCB) is of critical importance for an assessment of the onset and development of the Cordillera.

Aside from the sharp geological contrast across the C-S facies boundary, which separates the Paleozoic platformal carbonate sequences of the eastern Foreland Belt from the basinal chert and shale of the western Foreland Belt[24] (Fig. 1b), changes in physical properties (e.g., crustal/mantle seismic velocities[25–31], surface heat flow[32] and mantle electrical conductivity[33]) are well documented near the RMT. The crust and lithosphere also exhibit significant eastward thickening, by >10 km (ref. [25]) and >200

km (refs. [15,23,34]), respectively. However, the precise location and morphology of the CCB, especially at sub-crustal depths, remain speculative due to insufficient spatial sampling in previous geophysical surveys. Here we present updated geophysical constraints based on a decade (2006–2015) of broadband recordings from dense seismic arrays in western Canada. This dataset enables a higher resolution illumination of the 3D seismic P-velocity and S-velocity structures of the Cordilleran foreland region than previously available. By integrating seismic imaging with geodynamic calculations and surface geology, our study sheds new light on the mantle structures and dynamics near the CCB.

## Results

**Tomographic models**. Our finite-frequency body-wave tomographic models (see Methods section) show mantle velocity structures across the region and delineate contrasting low and high wave-speeds to the west and east of the RMT, respectively (Fig. 2). Beneath the southern Canadian Cordillera, negative velocities of −2.5% (−3%) relative to the reference model[36] for P (S) waves extend to 300 km depth. To the east, positive velocities of 2% (2.5%) of P (S) waves are present beneath the Alberta foreland basin[34]. The western margin of the cratonic lithosphere is a steeply dipping high-velocity structure juxtaposed to the west with pronounced low velocities beneath the Canadian Cordillera. This boundary (i.e., CCB) is defined by a high amplitude velocity gradient and shows significant spatial and geometrical complexities along the strike of the mountain belt (Fig. 2g–j).

**Cordillera–craton boundary**. The CCB provides a key structural constraint on Cordilleran assembly. We determine its location in our P-wave and S-wave models and three published tomographic studies[29,30,35] using the maximum horizontal velocity gradient (see Methods section), assuming the boundary marks a sharp lateral change in physical properties (e.g., temperature, composition and seismic velocity). The resulting location varies among different tomographic models and forms a narrow (<200 km) zone surrounding the RMT (Fig. 3a). Our P-wave and S-wave results both place the CCB at a maximum distance of 40–50 km west of the RMT at 150 km depth (Fig. 3a), with a pronounced westward dip (a minimum of ~10° from the vertical) between 49 and 52° N (see AA′ in Fig. 2g, i). The location and dip are robustly determined based on our synthetic tests, which show a small (<10 km) lateral uncertainty of the boundary location and a well-constrained boundary geometry in this region (Supplementary Figs. 7 and 8). The boundary lies directly beneath the RMT north of ~52° N (Fig. 3a), where its dip changes to sub-vertical and then east-dipping within a short (<50 km) distance (see BB′ in Fig. 2h, j). Farther north, the boundary merges into the northern Rocky Mountain Trench-Tintina Fault (RMT-TF) system at ~54° N (Fig. 3a). In this area, the geometry of the craton margin cannot be robustly determined due to reduced station density (see Supplementary Fig. 7).

The greatest velocity increase occurs within a 100 km distance from the CCB (Fig. 3a), showing maximum horizontal gradients of 4% and 3.5% per 100 km, respectively, for P and S velocities at 150 km depth (Fig. 3a). The shear-velocity value is consistent with the >3% gradient observed in this region in a recent continental-scale shear-velocity model[30]. As temperature is the dominant control on upper mantle seismic velocity[37], the observed velocity contrast across the CCB provides constraints on the temperature variation (see Methods section). At 150 km depth, the P (4.3%) and S (7.0%) velocity contrasts indicate a decrease of 200–300 °C from the Cordillera to craton (Fig. 3b). The low Cordilleran

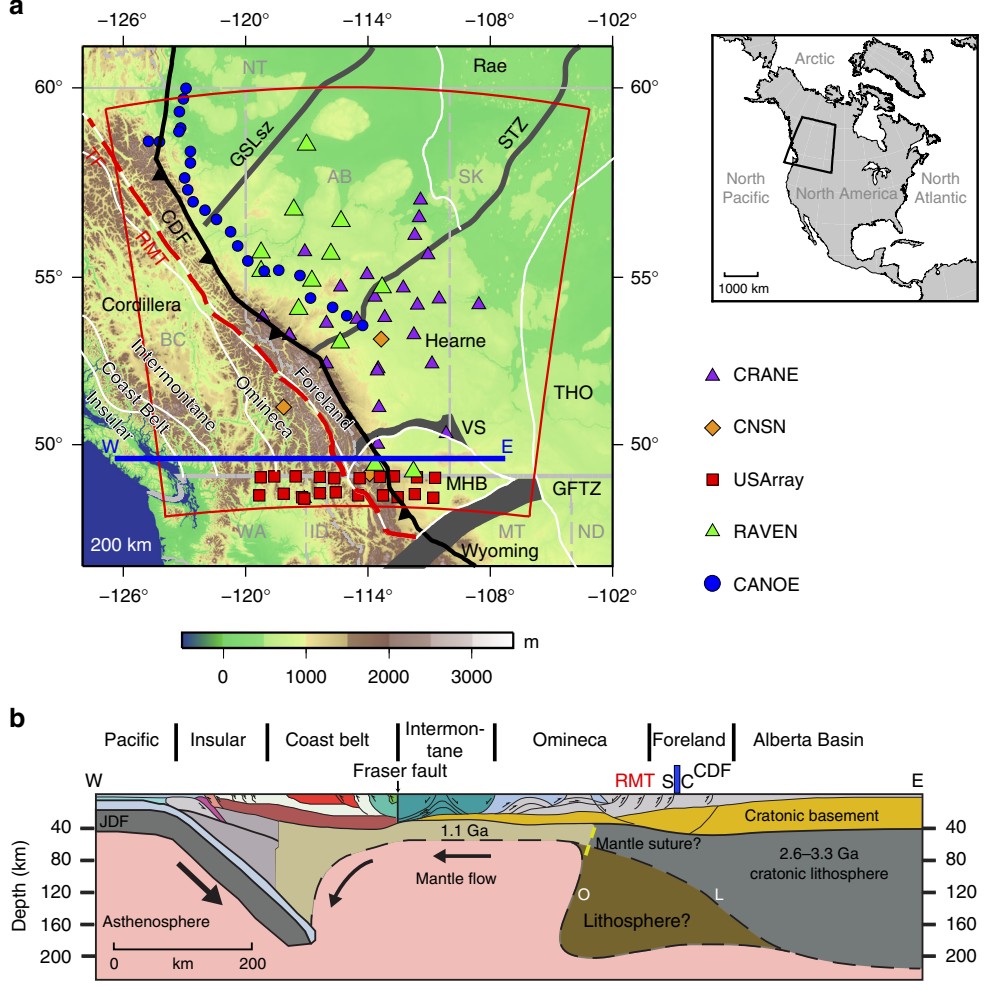

**Fig. 1** Tectonic setting of the Cordillera–craton transition region in western Canada. **a** Topographic relief map superimposed with station coverage. The boundaries of the tomographic model are denoted by the red polygon. Seismic stations are shown by various symbols. The thick gray lines represent the important structural discontinuities within the craton and the white lines indicate the major domain boundaries in southwestern Canada[19,20]. The black barbed and red dashed lines mark the location of the Cordilleran Deformation Front (CDF) and the Rocky Mountain Trench (RMT)-Tintina Fault (TF) system, respectively. The map area relative to North America is shown by the enclosed region in the regional map to the right. **b** A schematic cross-section showing the structural transition from the Cordillera to craton (location indicated by the blue line on the map). The blue bar marks the carbonate (C) to shale (S) facies boundary, the proposed Cordillera–craton surface suture. The thin dashed lines mark the interpreted lithosphere-asthenosphere boundary (LAB), with either a landward (L) or oceanward (O) dip inferred for accretionary and collisional models, respectively. Modified after ref. [21]. GFTZ, Great Falls Tectonic Zone; GSLsz, Great Slave Lake shear zone; MHB, Medicine Hat Block; STZ, Snowbird Tectonic Zone; THO, Trans-Hudson Orogen; VS, Vulcan Structure

velocities are consistent with a relatively wet, near-adiabatic mantle (1200–1350 °C). The temperatures of high-velocity cratonic lithosphere are 950–1100 °C, showing slightly higher values for a depleted composition; the craton velocities are insensitive to water content[38]. These temperatures are in agreement with earlier estimates based on surface heat flow, xenoliths, and seismic velocity[32,37,39,40], as well as the hypothesis that the thin Cordilleran lithosphere is maintained through small-scale convection of a hydrated mantle[39]. Based on geodynamic models, low craton temperatures in combination with a dry and moderately depleted composition are required to maintain a prominent (sharp and steep) lithospheric step at the craton edge for a minimum timescale of 100 Ma[41,42]. Specifically, the cratonic mantle lithosphere must be rheologically strong (5–50 times stronger than damp olivine, i.e., consistent with a dry composition) and chemically depleted (20–40 kg m$^{-3}$ less dense than primitive mantle)[42]. This intrinsically stable and potentially well-preserved mantle boundary thus provides critical temporal

constraints on the initiation and evolution of the Cordilleran orogen.

## Discussion
The location (subjacent to the cryptic suture in the foreland crust), westward-dipping geometry, and the large (>200 km) and sharp lithospheric thickness contrast at the CCB (see Fig. 2g–j) are key observations from our seismic models. They enable a new assessment of the tectonic paradigms (collisional or accretionary) for Cordillera evolution. In the collisional model, the Cordillera is a product of the Late Cretaceous collision between a pre-assembled (Triassic-Jurassic) ribbon continent and the North American continent[4,11,17], implying the existence of a collisional suture between allochthonous (i.e., Cordillera) and autochthonous (i.e., craton) mantle. Within this framework, the craton margin and CCB were established relatively recently (younger than 100 Ma) compared with the Late Devonian age suggested by

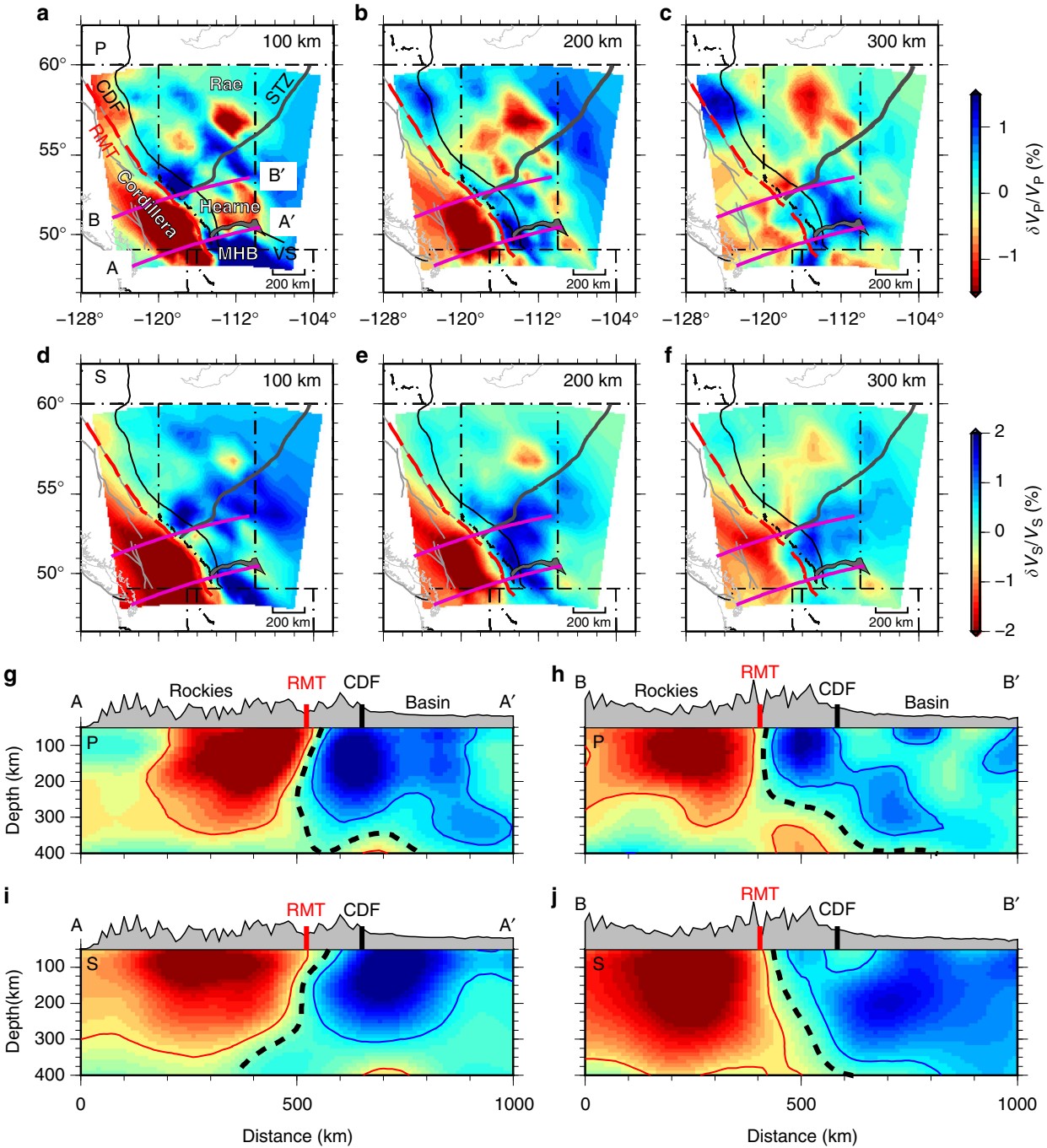

**Fig. 2** P-wave and S-wave velocity anomalies resolved from finite-frequency tomography. **a–c** P-wave velocities at 100, 200, and 300 km depths, respectively. **d–f** The same as **a–c** but for S-wave velocities. The locations of two velocity profiles are shown by the purple lines at 100 km depth. The red dashed line marks the Rocky Mountain Trench. **g, h** P-wave velocity anomalies along two profiles in the southern Canadian Cordillera. **i, j** The corresponding S-wave velocity variation along the two profiles. The southern (AA′) and northern (BB′) profiles intersect with the Rocky Mountain Trench at about 50° and 52° N, respectively. The locations of the Rocky Mountain Trench and Cordilleran Deformation Front are respectively marked by the red and black lines at the surface. The black dashed line indicates the zero percent velocity contour, which approximates the location of the Cordillera–craton boundary

the accretionary model[13,21]. Therefore, the collisional model infers a relatively short-term (<100 Ma) evolution (reworking) of the craton margin, providing an important temporal constraint on the preservation of the CCB, particularly its location and geometry.

Our seismic models show a well-defined westward-dipping CCB beneath the Cordilleran Foreland Belt. At 150 km depth, a robustly resolved depth range in our seismic images, the mantle

CCB is ~50 km west of the surface suture (C-S facies boundary east of the RMT) in the southern Cordilleran foreland. In this region, the amount of the Late Cretaceous and Paleocene shortening as accommodated by the Rocky Mountain thrust-and-fold belt is ~200 km (see ref. [43]). Following the shortening, the release of compressive stress near the thrust termination in the foreland reactivated the basal décollement, causing post-Eocene normal faulting in the southern RMT[44] and regional

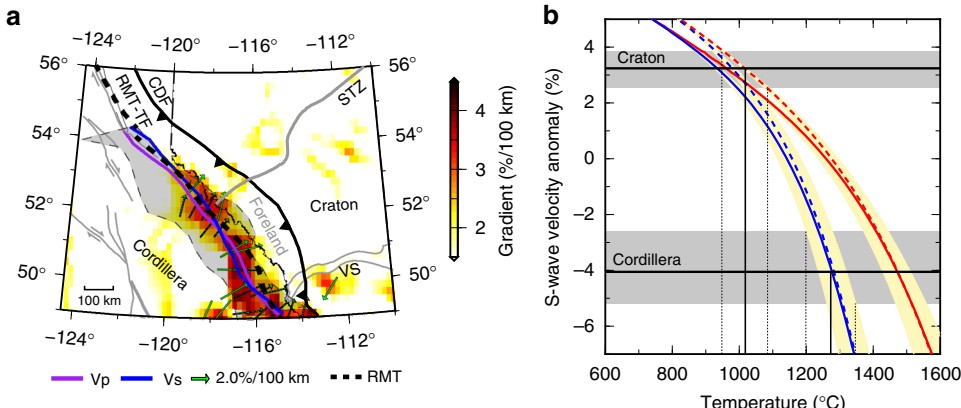

**Fig. 3** Seismic velocity gradient and temperature contrast at the Cordillera–craton boundary. **a** Seismic velocity gradient at 150 km depth from an averaged Vp and Vs velocity model scaled using Vp/Vs ratios in the reference model[36]. The Cordillera–craton boundary in the upper mantle determined from P-wave and S-wave models are shown by the purple and blue lines, respectively. The shaded region highlights the spatial variation in the Cordillera–craton boundary location measured from this study and three recent tomographic models[29,30,35] (Supplementary Fig. 5). The green arrows indicate the direction of the maximum velocity gradient near the transition boundary. Major faults are marked by the gray lines. **b** S-wave velocity-temperature (Vs-T) relationship at 150 km depth. The horizontal black lines show the respective average S velocities for the Cordillera and craton regions, with the corresponding standard deviations shaded in gray. The solid and dashed curves represent pyrolite (fertile) and dunite (depleted) compositions, respectively, in a dry (50 ppm H/Si; red) and wet (5000 ppm H/Si; blue) mantle. The yellow shaded regions show the variations of Vs-T curves for various frequencies (0.03–0.3 Hz) and grain sizes (0.03–3 cm)

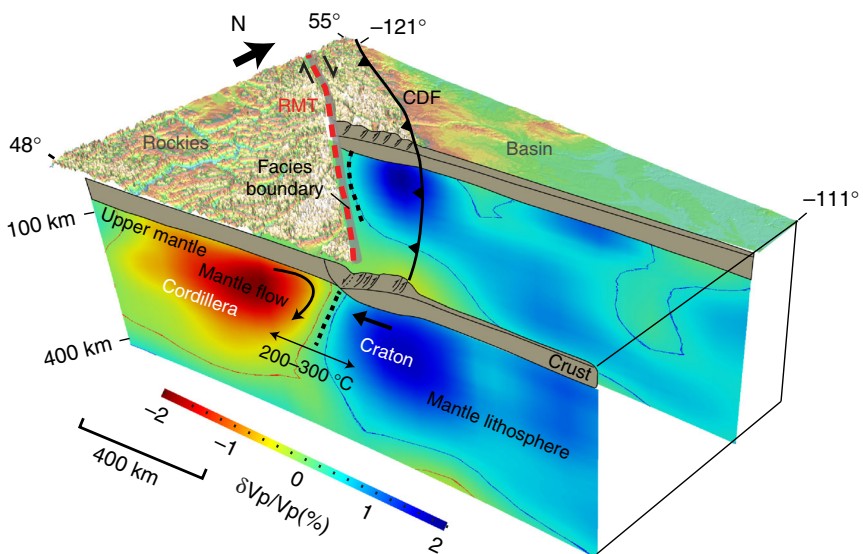

**Fig. 4** A 3-dimensional perspective of the Cordillera–craton boundary. The profile in the south is dominated by margin-perpendicular displacement whereas the northern profile is characterized by strike-slip motion. The Rocky Mountain Trench and Cordilleran Deformation Front are shown by the red dashed and black barbed lines, respectively. The surface facies boundary is shaded in gray and the mantle boundary is indicated by the black dashed line on the cross-section

extension of up to 25 km[45]. The close spatial association of the RMT with the CCB potentially suggests strong influences from minor reactivation of the CCB during extension. According to the collisional model, the compression stage is attributed to the convergence between the North American craton and the Cordillera[4]. During this protracted period of tectonic interaction (i.e., collision), the mantle CCB, which marks the collision front, moved continuously westwards as a consequence of the underthrusting of the leading edge of the craton[17] while the overlying crust carried the surface suture eastwards along the basal décollement of the thrust-and-fold belt[4]. The crustal extension partially restored the position of the surface suture relative to the stationary mantle suture, resulting in a net offset

of ~50 km between the two structures. The collision process provides a straightforward yet self-consistent interpretation of the observed westward-dipping CCB (a relic collision front; Fig. 4) and its spatial correlation with the surface suture in the southern Canadian Cordillera. Similar phenomena have been documented in orogenic belts of Qinling-Dabie in central China[46] and Trans-European Suture Zone[41]. North of 52° latitude, the RMT joins the TF in northern British Columbia and Yukon and is characterized by >400 km of Eocene dextral strike-slip displacement[47,48]. The transition from convergent to strike-slip motion coincides with the change in dip direction (i.e., westward to sub-vertical/eastward; Fig. 4), implying a dominant margin-parallel component of transpressive motion

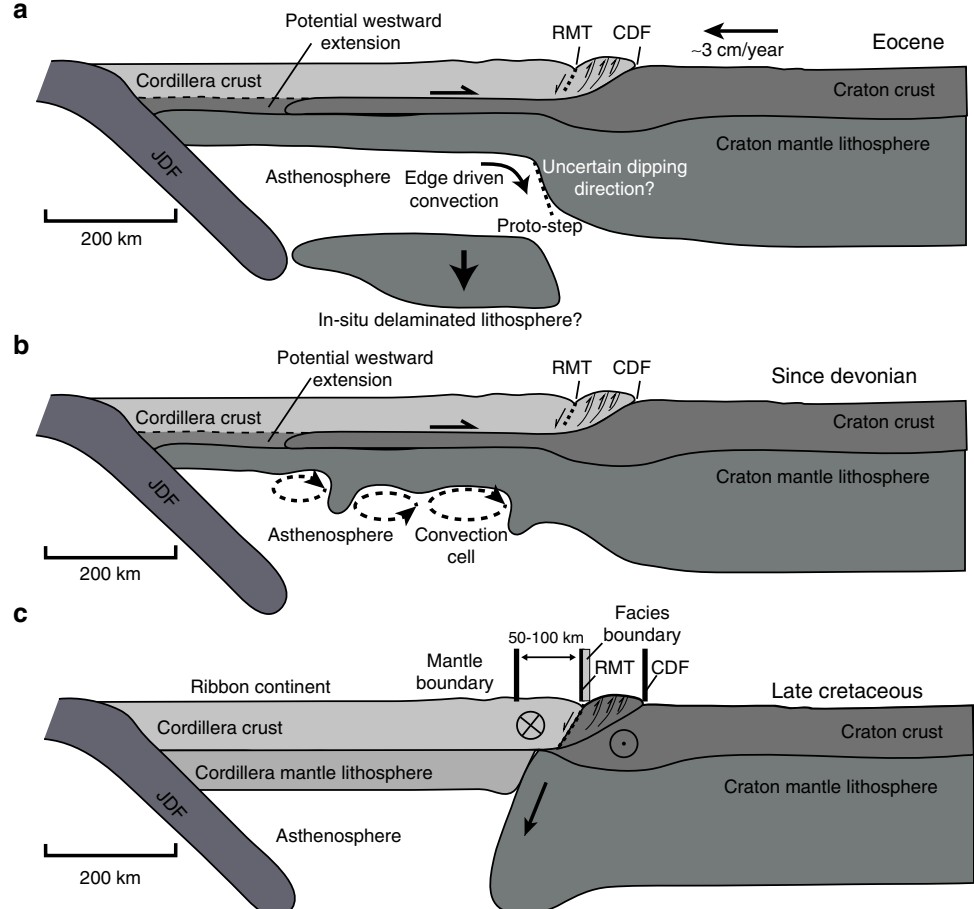

**Fig. 5** Three mechanisms for the formation of the Cordillera–craton boundary. In the accretionary hypothesis, the Cordillera–craton boundary formed as a destructive boundary through either **a** lithosphere delamination[23] or **b** viscous thermal erosion. In the delamination model (**a**), the absolute North American plate motion rate is obtained from refs. [50–52]. **c** A continental collision model that involves terminal suturing between a ribbon continent and the North America craton.

of the Cordillera relative to the craton in this region. This argument is corroborated by an overlapped surface suture and mantle boundary (Fig. 3a) and supports the interpretation of the TF as a lithosphere penetrating structure[49].

The interpretation of the CCB as a collisional boundary differs significantly from the views of the CCB in the accretionary hypothesis[13]. This model predicts that (1) the craton margin was established by at least the Late Devonian; (2) only the supra-crustal rocks of the exotic terranes were added to the North American margin; (3) the Intermontane Belt comprises the easternmost extent of accreted terranes and all crust farther east is North America[13,15]; (4) the lower crust and lithosphere beneath the Cordillera is a westward extension of the North American craton; and (5) the lithospheric mantle thins gradually to the west from the craton to Cordillera[13,21]. In this model, the evolution of the CCB has undergone at least two distinctive stages: the initial building of the Cordillera through the accretion of exotic terranes, followed by the lithosphere removal to create the sharp present-day craton boundary. Removal could be achieved through regional lithosphere-scale delamination[23] (Fig. 5a) and/or viscous thinning and thermal erosion of the Cordilleran lithosphere in a retro-arc setting[22,53,54] (Fig. 5b). In both cases, the formation of a steeply-dipping boundary underneath the Cordilleran foreland is closely associated with destructive processes. According to the delamination model (Fig. 5a), the boundary location is controlled by a proto-step beneath the RMT, which, in combination with the

edge-driven convection, jointly triggered removal[23]. This provides a possible explanation for the spatial affinity between the CCB and the RMT. However, the post-Eocene normal faults of the RMT region are younger than the suggested Eocene delamination event and a single large-scale delamination event does not account for the observed diverse geometry of the CCB along the strike of the orogen (Figs. 2, 4). Additionally, the interpreted present-day position of the delaminated lithosphere below its point of origin[23] (i.e., west of the CCB and immediately beneath the Cordillera; Fig. 5a) is difficult to reconcile with the continual westward motion of North America. The average absolute North American plate motion rate of ~3 cm per year since the proposed delamination event at 55 Ma (see ref. [50–52]) would place the detached block ~1500 km to the east relative to the overlying continent.

A more plausible destructive mechanism may involve smaller-scale viscous/thermal erosion that removes a significant amount of mantle lithosphere from beneath the Cordillera (Fig. 5b). However, the accretionary model predicts that the sub-Cordilleran mantle lithosphere is composed of dry and buoyant rocks that are intrinsically resistive to erosion (Fig. 5b). Therefore, strength reduction is needed to promote thinning, possibly through continent rifting followed by refertilization and/or slab dehydration above an east-dipping subduction zone[39,54]. In this interpretation, the CCB marks the eastern limit of the erosion front and its present-day geometry (e.g., Fig. 2) is evidence of a

sharp rheological boundary. This boundary has either persisted as a long-lived (i.e., since the Devonian) rheological difference or that the current (foreland) position of the CCB reflects a snapshot of an eastward-migrating craton margin[54]. It is not clear that either of these scenarios can create the observed large gradients in seismic velocity from the Cordillera to the North American craton.

For the southern Canadian Cordillera, the new seismic observations are more compatible with the collisional model that provides a self-consistent mechanism to explain (1) the steep and well-preserved west-dipping geometry—a young (<100 Ma) collision front; (2) the sharp velocity, temperature and lithospheric thickness contrasts indicating a boundary separating two distinct lithospheres; and (3) the excellent spatial correlation (and offset) with the cryptic surface suture (Fig. 5c). Collectively, these spatio-temporal constraints on the CCB could signify periods of ribbon continent formation and its later collision to the autochthonous domains (i.e., North American craton; see ref. [4]). The collisional process predicts a crustal suture in the foreland, the identification of which will be critical for substantiating this hypothesis and requires high-resolution seismic imaging of the crustal structures.

Although our interpretation of a collisional suture beneath the foreland of the Cordillera is based on a study of the southern Canadian Rocky Mountain region, our model implies that the mantle seismic structure (i.e., lithospheric suture) extends southwards into the United States (Fig. 2a–f). This is corroborated by the geological observations across international border, where continuous structures, stratigraphy and geological belts have been reported. Specific examples[4,6] of continuity include, from west to east, Triassic/Jurassic magmatic arc sequences (Quesnellia in the north and Wallowa and Olds Ferry to the south); the Mesoproterozoic and Neoproterozoic Belt-Purcell and Windermere supergroups; mid-Cretaceous (120–90 Ma) granitoid plutons (Omineca Magmatic Belt in the north and Idaho batholith to the south) that intrude the Precambrian and younger strata; and Jurassic to Paleocene, east-verging fold-and-thrust belt structures (Columbian and Rocky Mountain in the north, versus Sevier and Laramide in the south). Although cryptic, the surface trace, and its mantle counterpart, of the proposed suture likely continues southwards within the foreland fold-and-thrust belt, east of and structurally beneath the Belt-Purcell sequence[4,11,17].

Suture zones place crucial constraints on continental assembly, although the recognition of distinctive plate boundaries at shallow (e.g., surface ophiolite belts) and deep (e.g., lithospheric fault zones) structural levels is not trivial[18]. Our analysis of the southern Canadian Cordillera combines deep structural constraints from seismic tomography with surface geology to shed new light on the close relationship between a surface cryptic suture and its upper mantle expression (see Figs. 4, 5c). The sharp structural and temperature gradients associated with the CCB could be associated with a stable craton margin established during the collision of a ribbon continent (Cordillera) with the North American craton in the Late Cretaceous, although other scenarios (e.g., thermal/viscous erosion) cannot be fully excluded. For example, the formation of the CCB via gravitational thinning of the Cordilleran lithosphere based on the accretionary model provides an alternate interpretation; further analyses would be needed to understand the potential thermal/dynamical processes that create the sharp gradient near the CCB. An integrated approach, as used in our study, is paramount to deciphering the style[9] and initiation[55] of orogenesis, and provides a testable tectonic framework for future investigations. As more data and examples become available in other orogens, new insights into the dynamics of the crust and mantle during orogenesis and continental growth can be gained.

## Methods

**Finite-frequency tomography.** The P-wave dataset consists of 23,123 teleseismic arrival times from 1761 earthquakes and the corresponding S-wave dataset includes 17,253 arrivals from 1263 earthquakes (Supplementary Fig. 1). P phases are measured from vertical-component seismograms within frequency ranges of 0.03–0.125 (low) and 0.3–2.0 Hz (high) to minimize a noise peak at 0.2 Hz and to take advantage of the wide bandwidth of the earthquake signals. The corresponding S waves, measured from the tangential component, are filtered at low and high frequencies of 0.03–0.1 and 0.1–0.2 Hz, respectively. Relative travel times among all stations recording the same event are measured using the multichannel cross-correlation method[56]. The final relative travel-time residuals are computed by subtracting the demeaned theoretical relative travel times from those observed, which generally follow normal (Gaussian) distributions with a respective standard deviations of 0.4 and 1.3 s for P and S phases (Supplementary Fig. 1).

The region of study is characterized by large topographic reliefs and crustal contrast between the Canadian Rockies and the Alberta basin, which contributes to the travel time fluctuations across the recording array. We apply topographic and crustal corrections to minimize these effects caused by the shallow structures. The former term equals to the travel time within the crustal segment above sea level (i.e., extra topography). The latter term is defined by the travel-time difference between the observed (CRUST1.0[57]) and theoretical (AK135 continent model[36]) values and is calculated by tracing a ray through the crustal layers in each of these two models. Large values are observed along the foothills of the Rockies (Supplementary Fig. 2), where a thick (~50 km) crust exists in response to the load of supracrustal rocks of the foreland thrust-and-fold belt[43]. The final correction at a station is made by subtracting the topographic and crustal correction terms from the measured relative travel-time residuals. The resulting time-corrected data show a clear east (positive)-west (negative) contrast that generally follows the Cordilleran Deformation Front (Supplementary Fig. 2).

Finite-frequency theory[58,59] forms the basis of our travel-time tomography scheme, which relates observed travel-time measurements to slowness structures through the following equation:

$$\delta t = \iiint_\oplus K(x)\delta s(x)d^3x, \qquad (1)$$

where $K(x)$ is the Fréchet derivative (i.e., sensitivity kernel) that maps the slowness perturbation $\delta s$ at a point x within model volume $\oplus$ to relative travel-time residual $\delta t$. The kernel is computed using the Born forward scattering theory in combination with paraxial ray approximation[58,59], which properly considers the effects of wavefront healing and diffraction on seismic wave propagation (and hence travel-time shifts). Our region of study is parameterized into a spherical grid covering an area of $12 \times 12$ degrees and extends 800 km in depth. The number of nodes is 33 along each direction, resulting in a grid size of approximately 40, 40, and 25 km in latitude, longitude, and depth. The model parameters can be solved by formulating Eq. (1) into a concise matrix form

$$\mathbf{d} = \mathbf{Gm}, \qquad (2)$$

where $\mathbf{d}$ is the data vector that contains M (23,123 for P and 17,253 for S) relative travel-time residuals and $\mathbf{m}$ is the model vector that contains N ($33 \times 33 \times 33 = 35,937$) slowness parameters. The corresponding inversion kernel ($\mathbf{G}$) is then a M × N matrix that defines the sensitivity of the datum ($\mathbf{d}$) to slowness perturbation ($\mathbf{m}$). Instead of solving Eq. (2) directly in a grid-based parameterization, we solve P and S velocities independently and transform the model vector and inversion kernel into the wavelet domain. We then seek a damped least-squares (DLS) solution for wavelet coefficients corresponding to each wavelet basis or hierarchical scale. This approach allows a data-adaptive scheme of non-stationary regularization, thus yielding spatially varying resolution in the resulting model. More details on finite-frequency theory and multiscale parameterization can be found in ref. [60].

**Boundary determination.** The interpretations of tectonic structures in tomographic images are often based on visual perceptions of colors, which typically associate targeting geological structures (e.g., slab, continental lithosphere, and hot plume) to anomalies confined within a specific velocity contour. This may lead to potential interpretation biases (e.g., underestimate or overestimate of anomalies) that result from a subjective choice of contour value. In addition, even tomographic models of the same region can exhibit considerable variations due to different data type/coverage, model parameterization, as well as inversion and damping schemes (Supplementary Fig. 3). As a result, the comparisons of key structures (e.g., the CCB in this study) from different models often lack a systematic criterion and remain largely qualitative.

To quantitatively determine the location of the transition from tomographic models, we use a maximum velocity gradient approach that is insensitive to the background velocity (i.e., determined by relative velocity perturbation). We compute the horizontal velocities in a depth range of 100–200 km along a series of parallel cross-sections perpendicular to the strike of the Rockies (Supplementary Fig. 4a). Assuming that the CCB marks a sharp change in physical parameters (e.g., velocities, densities, and temperatures), we choose the point of maximum velocity gradient as the optimal boundary location at each depth (Supplementary Fig. 4b). This criterion has been applied to the determination of depth of the LAB[61,62]. To

avoid the spurious maxima caused by noisy data (i.e., high model roughness), we fit the velocity with lower-degree polynomials, which are degree 3 for S and degree 5 for P while considering higher frequency (i.e., shorter wavelength) nature of the latter phase. To capture the trend of local velocity variation along the profile, we use a ~500 km wide sliding window during the fitting process, which approximates the wave-length of the slow to fast velocity transition (e.g., from 250–750 km in Supplementary Fig. 4b). The final boundary location is calculated by averaging all boundary points (Supplementary Fig. 4c). The same method is applied to five tomographic models for boundary determination (Supplementary Fig. 5).

**Model resolution**. We perform checkerboard tests to evaluate the resolution of our P and S velocity models. The input structures consist of alternating positive and negative Gaussian-shaped anomalies with maximum amplitudes of 3 and 5% for P and S velocities (Supplementary Fig. 6), respectively. Each anomaly spans 7 nodes in three directions, forming a volume of ~$240 \times 240 \times 150$ km$^3$. Synthetic travel times are calculated based on actual event-station geometries, and random errors with a standard deviation of 0.06 and 0.16 s, resembling the uncertainties in the observed travel times, are subsequently added to P and S data, respectively. The same parameterization and regularization schemes used in the actual inversion are adopted during the inversion of the synthetic data. The output of P velocity model successfully resolves the checkerboard in central-southern Alberta and south-eastern BC with 60% recovery of the input amplitudes. The S-wave model shows slightly lower degree (40%) of amplitude recovery than P. In both models, the lateral resolutions degrade at shallow depths (e.g., 100 km), where the converging rays cause reduced sensitivity near recording stations. The resolutions are highest in the Cordillera–craton transition region along the Rockies, where our data are sufficient to resolve a P velocity anomaly with respective lateral and vertical dimensions of 100 and 150 km. On the other hand, the S model is subjected to more severe vertical smearing effects, hence the minimum vertical scale resolvable is approximately 50 km less than that of the P model.

We conducted hypothetical tests to evaluate the uncertainty in the determined boundary location. The first test includes a gentle westward-dipping boundary separating low (−3.5%) and high (2.5%) P velocity anomalies (Supplementary Fig. 7a–d). The model outputs show excellent recovery of the boundary location with 10–30% underestimate of peak input velocities in the south and 40–60% in the north. The location uncertainty is small with the maximum discrepancy (20–30 km) observed in the north (Supplementary Fig. 8). The corresponding test for the S model utilizes input low and high velocities of −4.5% and 3.5%, respectively. The boundary is well defined in the output model with a difference of 6–30 km compared to the input (Supplementary Fig. 8). The second group of tests adopts the same model input as the first test case except for a vertical boundary (Supplementary Fig. 7e–h). The output models show virtually the same degree of recovery in boundary location, which suggest a relatively minor effect of boundary geometry on the determined location (Supplementary Fig. 8).

The geometry and sharpness of the boundary provide important structural constraints to the Cordilleran tectonics. We further examine and discuss the resolvability of our data to these parameters. For a westward-dipping boundary (Supplementary Fig. 7a), the geometry is well-constrained in the south, but the degree of recovery degrades towards the north, where the boundary is steeper and more diffuse compared with the input. For a vertical boundary (Supplementary Fig. 7c), geometry and sharpness of the boundary are both well recovered in the south between 48 and 52° N, whereas an artificial westward dip is observed in the north. By comparing the results of these two tests, we conclude that (1) the observed boundary characteristics (sharpness and dip) is robustly determined in the south and (2) the resolution degrades towards north (above ~52° N) and the dip may be artificially skewed. Hence, caution needs to be exercised when interpreting the boundary geometry in this region. For the S model (Supplementary Fig. 9), our tests show a more severe underestimate of the dip compared to the P model and the geometry cannot be reliably determined in the north. These test results suggest that P waves are more sensitive to boundary geometry compared to S waves. In our model, the dip of the boundary transitions from westward to eastward dip at ~52° N, which we determine to be a reliable observation since the model recoveries are satisfactory on both P and S models; more importantly, we find that no artificial eastward dip occurs at this latitude in all test cases.

We further examine the effect of separation distance (i.e., sharpness) between low- and high-velocity anomalies on the recovered boundary geometry. We use the same dipping structures as those from earlier tests, but increase the separation distance to 50 km (Supplementary Fig. 10). The output model is again able to resolve the geometry and gradient of the input boundary. For the final test case, the separation distance is increased to 100 km, and the inversion only recovers the gradient but fails to resolve the dip of the boundary. In summary, these resolution tests suggest that our data are sufficient to distinguish a relatively sharp (<50 km) boundary with a dipping geometry, whereas the boundary cannot be fully resolved if the transition occurs over a relatively large distance (e.g., 100 km). A corollary of this test is that a sharp (within 50 km distance) boundary must be present in the southern Canadian Cordillera, where a steep westward dip is clearly defined in our article.

**Temperature calculation**. Insufficient data coverage and smoothness constraints (i.e., damping) in the inversion are known issues that can weaken the amplitudes of the recovered seismic anomalies. We consider these effects on the measured velocity contrasts across the CCB and apply correction factors to compensate for the amplitude reduction. Correction factors are calculated from the percent of underestimate by comparing the input and output peak velocities of the models used in the hypothetical tests. The corresponding uncertainties are derived from the standard deviation of the results of different damping values near the turning point of the trade-off curve. The corrected values show larger and nearly constant velocity increase across the transition boundary between 49 and 54° N (Supplementary Fig. 11a, b), which agree with the observations from a recent tomographic model[30].

We compute the temperature using the seismic velocity after correcting for the underestimate during the inversion. We limit our calculation to the south of 52° N, where the model resolution is the highest. We convert the velocity perturbations to absolute velocities based on AK135 reference model[36] considering model dependencies in travel time prediction, ray-tracing and kernel construction. We follow the approach of ref. [38] to map tomographic velocity variations to temperature. Anharmonic P and S velocities as a function of composition, pressure and temperature are obtained using Perple_X[63]. These are then corrected for anelastic effects based on experimentally derived parameters (Eq. 4 in ref. [63]), using a grain size of 1 cm and frequency of 0.1 Hz; variations of 0.3–3 cm and 0.03–0.3 Hz are considered. Calculations are carried out for the primitive and depleted mantle, using a pyrolite[64] and dunite[65] composition, respectively, and for water contents of 50 ppm H/Si (dry) and 5000 ppm H/Si (wet). Percentage velocity anomalies are relative to the AK135 velocity model, consistent with the tomographic inversion approach. The resulting temperature profiles from P (Supplementary Fig. 11c) and S velocities (Fig. 3b) yield consistent temperature differences of 200–300 °C between the Cordillera and craton.

## Data availability

Seismic data for USArray, RAVEN, and CANOE networks are provided by IRIS Data Management Center (http://ds.iris.edu/ds/nodes/dmc/). Seismic data for CNSN network can be requested from Canadian National Data Center (http://www.earthquakescanada.nrcan.gc.ca/stndon/CNDC/index-en.php). Traveltime data could be accessed through website https://sites.google.com/a/ualberta.ca/seisworld/data.

## Code availability

The codes of tomographic imaging and velocity gradient analysis are available from the corresponding author upon reasonable request.

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

## Acknowledgements

We thank Martyn Unsworth for comments; and Global Seismology Group at the University of Alberta for field support. Y.J.G. was supported by funds from Future Energy Systems at the University of Alberta. Y.J.G., C.A.C., S.T.J. and P.A. were supported by Discovery Grants from Natural Sciences and Engineering Research Council of Canada (NSERC). S.H. was supported by the Ministry of Science and Technology of Taiwan (grant 107-2116-M-002-020).

## Author contributions

Y.C., Y.J.G. and S.H. contributed to the body-wave tomography analysis. C.A.C. conducted the geodynamic calculation and interpreted the results. S.T.J. provided geological background and contributed to the ribbon continent hypothesis. A.J.S. and P.A. contributed to velocity gradient analysis. Y.C. primarily wrote the manuscript, with substantial input from Y.J.G. and additional input from all co-authors.

## Additional information

**Competing interests:** The authors declare no competing interests.

