## [Peer Review File · Nature Communications]

Reviewers' comments:

Reviewer #1 (Remarks to the Author):

This is my second review of the manuscript by Chen et al. for Nature publications. As such, I will duplicate the parts of my first review that remain relevant.

Summary of Key Results

The authors use recently available high-density seismic arrays, through estimation of P- and S-wave velocity anomalies and gradients, to better define the 3D boundary between lithosphere underlying the craton and lithosphere underlying the Cordilleran orogen. The resolution of the geometry of the boundary, in particular its local west dip, are used to argue that this represents a relict Cretaceous collisional boundary between North America and a ribbon continent. Geologic arguments for the existence of the collision boundary in the upper crust are combined with geophysical observations for its deeper structure.

Originality and significance

The differentiation between collisional and accretionary boundaries in ancient orogens is a question of first order importance. The accretionary and/or non-collisional models are the most widely accepted viewpoints of orogenesis of the North American Cordillera, and a provocative model of Cretaceous collision of a ribbon continent has been published and advocated for by few researchers. Thus, geophysical evidence that can bear on the nature of Cordilleran orogenesis is important. The authors have well described a fairly sharp western boundary in the North American craton, and a significant change in thickness across this boundary. They then interpret the steeply west dipping boundary as a relict Cretaceous collisional boundary.

The authors now address three alternative hypotheses for generating an abrupt step in the lithospheric thickness across what they interpret as a suture. This, in my opinion, is a more balanced approach to the interpretative process and adds to the credibility of their favored interpretation. Please note that on the newly added figure showing the three alternative hypotheses, there are several errors in spelling (potential, delaminated).

Geodynamic Models

As the crust of the Rocky Mountain fold-thrust belt underwent a minimum of 220 km of shortening, mass balance considerations require a commensurate amount of shortening of the mantle lithosphere. And yet, the mantle lithosphere under the Canadian Cordillera is quite thin. (An alternative is that shortening in the fold-thrust belt is balanced by accretion of terranes at the plate margin following the concept of orogenic float. Orogenic float does not provide a viable mechanism for mass balance in the US Cordillera, so I will, for the sake of argument, discount it for the Canadian Cordillera). This then requires a mechanism for loss of mantle lithosphere that was thickened to achieve the now relatively thin state, with ablative subduction, vigorous thermal or viscous thinning, or delamination as possibilities. Thus, what may be imaged is the geometry of the destructive feature, not a collisional boundary.

The authors provided the following response to the comment above:

"We thank Reviewer #1 for pointing out the potential mass balance problem associated with shortening of the Cordilleran foreland fold-and-thrust belts. However, we point out that a significant mantle thickening is not required to accommodate 220 km of shortening in the thin-skinned (2 to 4 km stratigraphic succession) fold-and-thrust belt. A simple geometric calculation shows only 10 km of distributed shortening within a 70 km thick mantle section balances observed shortening in the fold-and-thrust belt. On the scale of the Cordilleran orogen, such a small amount of shortening would not result in any significant mantle thickening".

I find this logic to be flawed. For strain compatibility, and mass balance, each layer in the lithosphere has to experience the same amount of strain. If the upper crust experiences 220 km of

shortening, then any arbitrary layer within the mantle lithosphere will also need to experience 220 km of shortening to maintain strain compatibility. The attempt at area balance in the response is incorrect.

Regarding issues of the preservation of a lithospheric step at the craton edge, arguments are but forward for the requirement of different mantle lithospheres on either side of the boundary to both generate and preserve this prominent step. This is actually a critical point and is perhaps the most important interpretation in differentiating between the 3 different hypotheses. Differences in rheology, due to differences in chemical depletion and water content are discussed within the framework of collision of two different lithospheres. However, is it not possible to generate differences in rheologic (and geophysical) response simply due to the differences in geologic history and lithospheric modification between the Cordillera and the craton? The Cordillera experienced a geologic history of rifting followed by a prolonged location in a suprasubduction zone environment. The lithosphere at the craton margin experienced rifting and attenuation during the breakup of Rodinia -- with all the associated processes of lithospheric modification -- and then protracted modification by processes associated with subduction including interaction with volatiles from dehydrating slabs, convective eddies ... etc. Can the hypothesis that the rheological differences result from these processes superimposed on the same lithosphere be falsified?

Geological Constraints

Application of the hypothesis of a ribbon continent colliding with the North America craton through westward subduction faces challenges in the geologic constraints. As mentioned previously, I am not an expert on the geology of southern Canada but I have the following concerns with the application of the hypothesis south of the Canadian-US border (and I note that section A-A' spans the border). The proposed suture would separate the Windermere Group and its US equivalents from the carbonate passive margin, placing the Windermere within the posited ribbon continent. In the US Cordilleran fold-thrust belt, progressive changes in sedimentary facies and thicknesses can be traced from the craton into the western passive margin between successive thrust sheets, showing a coherent, westward-thickening passive margin sedimentary prism, and leaving no room for juxtaposition of a ribbon continent. Even within single thrust sheets – such as the Willard sheet in Utah and Wyoming – these E to W progressive changes related to the passive margin can be seen, with westward thickening strata and the continuation of “cratonal” Paleozoic strata depositionally overlying the wedge-shaped Neoproterozoic to Early Cambrian Windermere-affinity sedimentary prism. Two of many examples would be the thrust belt near Las Vegas (see Burchfiel et al., 1974 GSAB), and the Idaho-Utah-Wyoming salient of the thrust belt (see Yonkee and Weil, 2015, ESR). The second inconsistency is in the timing of emplacement of the thrust sheets with Windermere-related strata. Two thrusts that emplace the Windermere-related rocks include the Wheeler Pass thrust in the Spring Mountains in Nevada and the Willard thrust in Utah. The former is a Late Jurassic structure (Giallorenzo et al., 2018, GSAB) and the later initiated slip at ca. 130 Ma (Eleogram, 2014; Gentry, 2016, and many others), and there is no evidence that they were emplaced in a Late Cretaceous collisional event. With regard to the Belt-Purcell Group, the Belt sits depositionally on top of Proterozoic crystalline basement in SW Montana (LaHood facies), and also occurs east of the thrust belt in the Little Belt Mountains; yes, parts of the Belt Supergroup experienced major translation during Mesozoic to early Tertiary thrusting, but the entire basin is not allochthonous. The point here is that it is very difficult to apply the ribbon continent collision model as advocated by Hildebrand to the US Cordillera. As to your mention of Paul Hoffman's support for this concept, please note that Paul Hoffman, while an outstanding tectonicist, has no track record of working on the Mesozoic Cordillera of the US. I also note that Paul has played a major role in getting Hildebrand's contributions published by the Geological Society of America, so of course he is in support of geophysical arguments that support this model.

Data & methodology: validity of approach, quality of data, quality of presentation

I am not a geophysicist and so I cannot evaluate these issues. However, following my earlier comment regarding better addressing why this data is unique, different, an improvement, etc, over the prior geophysical data sets that have imaged across this boundary, I think the authors

have well addressed these issues.

Conclusions: robustness, validity, reliability

The authors have delineated a clear boundary between thicker, seismically fast lithosphere, and thinner, low-velocity lithosphere to the west. Although I am not a geophysicist, the methods, documentation, and resolution of this feature seems very robust. I am less convinced of the interpretation of this boundary. The boundary is slightly west dipping in profile A-A', but is steep to east-dipping in profile B-B', and yet the authors embrace the west-dipping geometry. The geologic predictions as applied to the US Cordillera do not match the geologic record, despite the numerous publications by Hildebrand. If the advocated model would still be valid if it did not make predictions south of the international border and be specifically relevant to the Canadian Cordillera, then I do not have the expertise to assess this validity and would recommend a review be solicited from someone like Ray Price. In either case, the authors do provide geophysical evidence that permits a collisional hypothesis, and although I am not convinced of the validity of this hypothesis to the western Cordillera, it is provocative and will provide a platform for future work in testing the ribbon continent collision model

Reviewer #2 (Remarks to the Author):

The authors have satisfactorily answered the comments of all reviewers and their paper, which provides new seismological findings of importance for any interpretation of Cordilleran tectonic development in western North America, should be accepted without further revision for publication in Nature Communications.

I have discussed the lithostratigraphic correlations said to tie the outboard terranes with cratonic Laurentia with the geologists responsible. Their interpretations may be correct but alternative interpretations are also possible: critical areas are heavily forested with poor bedrock exposure. The basic mapping was done decades ago when alternative interpretations were seldom considered.

Cordilleran geologists who two decades ago dismissed without explanation a mountain of paleomagnetic evidence should not now be able to block soundly-based (pun unintended) geophysical data.

The tectonic interpretation offered by the authors may also be non-unique, but it is the most parsimonious one in light of existing data. The paper is also valuable for exposing the weaknesses and contradictions in alternative interpretations involving cratonward-dipping subduction beneath western Laurentia since the mid-Mesozoic.

The westward-dipping Cretaceous subduction-collision model is not without its own problems (e.g., dynamic subsidence and uplift history of the Western Interior basin). These lie outside the scope of the present paper, which has already been delayed for over half a year since my earlier review.

Reviewer #3 (Remarks to the Author):

I reviewed a previous version of this manuscript (as Reviewer #3) and believe it has been improved especially in terms of discussing and evaluating previous model (such as Bao et al.) and a more justified interpretation of the tomographic images, but there are still a few issues that need

to be addressed.

I did note that Currie is a co-author which is why I was surprised that her prior work was not discussed in more detail. I am happy to see they are now.

So what is really the difference between the CCB and RMT? As it is written I don't see the main distinction. Can they be the same? Or are they fundamentally different tectonic features?

Line 84. Delete "unique"

Line 88-89. Delete ", and thus sheds new light on the nature of the southern Canadian Cordillera"

Line 94. Ref 32? Wrong reference?

Line 101. replace "presumed" with "interpreted"

Lines 120-122 and Figure 2. The color scale in Figure 2 does not extend to 2.5% (or 3%). The colors are saturated or you are citing the wrong values in one place. And in the text the velocity perturbations should be a negative or more specially say a reduction.

Line 126. What subsurface extension? Inferred from what data or study? This is unclear.

Line 129. Delete "the focus of this student"

Figure 4. If this is really the tomographic model presented along those slices (which I assume it is) and not just a cartoon, then why not use exactly the same color scale and range as you do in Figure 2? Are the two profiles in the same position as those in Fig2B? You still are using the term "subduction" in the figure which is fundamentally incorrect.

Figure 4 (cont). The northern profile in the 3D figure has a low velocity subhorizontal feature around 200 km depth. this is not present in the cross sections in Figure 2B. What is this? Why haven't you addressed this? I see it in the depth slices in Figure 2A.

I would remove the Hildenbrand and Whalen reference as it doesn't really help your story. It is wild as I mentioned previously. And on a related point, now that you have toned down the "subduction" interpretation of the craton I don't think the westward subduction references or discussion really adds to your interpretation. The ribbon continent collision idea needs to be better supported in the first half of the paper. Much of the text still remains from the previous version that supports the westward subduction idea that you have now more or less discarded.

Response to reviewers:

This document presents our point-by-point response to reviewers' comments. The original comments are in ***Bold Italics*** and call-outs to the manuscript are shown in *Italics*.

The constructive comments from all reviewers from the last round have led to an expanded discussion of various hypothetical models in light of our new geophysical data. We believe that the reorganization and expanded discussion have greatly strengthened the arguments and improved the balance of our paper. Through careful assessment of three (one collisional and two destructive) models, we suggest that the spatiotemporal characteristics of the Cordillera-Craton boundary (CCB) is most compatible with the predictions of the collisional model while the two non-collisional mechanisms require additional processes/factors to facilitate the initiation (in case of delamination) or termination (in case of erosion) of the boundary.

We echo with Reviewer #1's comments on the originality and significance of this study and would like to reiterate the overarching goal of this study -- ***providing new geophysical constraints on the seismic and thermal characteristics of the CCB***. These new data offer an opportunity to assess the competing orogenic models of the North American Cordillera. Our study represents so far the best resolution available for the southern Canadian Cordillera. We show that the CCB has been, to a certain extent, misrepresented in earlier studies in terms of its location and geometry. This has implications for the interpretation of the geological evolution of this region. Whether the CCB is a collisional or non-collisional structure, the conventional ideas regarding the nature and the dynamical processes should be revisited in light of the new data. We believe that this opens a new window to future Cordilleran studies.

This point also echoes Reviewer #1's later comment that "*it is provocative and will provide a platform for future work in testing the ribbon continent collision model*". Indeed, our intention is to ***provide a testable tectonic platform that promotes, rather than terminates, the discussion over the long-debated questions about the tectonic style of the Cordilleran orogenesis***. In fact, there has been a growing research interest in the Cordilleran tectonics as evidenced from the publication record from this year alone. For example, *Zaporozan et al.* (2018) and *McLellan et al.* (2018) conducted surface wave tomography of the Cordillera-Craton transition region and *Polat et al.* (2018) provided new xenolith isotopic evidence on the complex nature of the Archean sub-Cordilleran mantle lithosphere.

In this revised version we carefully consider the additional comments made by Reviewer #1 and Reviewer #3 and have made the following major changes.

1. We revised the abstract to focus more on presenting the facts (observations) and provide a more objective opening. As per Reviewer #3's comments, we removed expressions such as "the collisional front" and "westward subduction".

2. We added a discussion about the destructive processes that could initiate (or maintain) the compositional/rheological difference between the Cordillera and Craton, an important factor for preserving the CCB.
3. We added discussions of model prediction. We suggest future research opportunities for testing the collisional model, which include locating the crustal suture based on integrated geophysical and geological studies.

We have also made a number of smaller changes and our detailed replies to the reviewer comments are given below. Over all, after two rounds of thought provoking comments from all reviewers, we believe that the structure of our paper is much more balanced and the discussion is more inclusive. We sincerely hope these changes to manuscript serve to clarify the scientific contribution of our work. We also hope that the readers find this paper appealing and helpful for understanding the current challenges and could inspire future Cordilleran studies.

Reviewer #1 (Remarks to the Author):

This is my second review of the manuscript by Chen et al. for Nature publications. As such, I will duplicate the parts of my first review that remain relevant.

Summary of Key Results

The authors use recently available high-density seismic arrays, through estimation of P- and S-wave velocity anomalies and gradients, to better define the 3D boundary between lithosphere underlying the craton and lithosphere underlying the Cordilleran orogen. The resolution of the geometry of the boundary, in particular its local west dip, are used to argue that this represents a relict Cretaceous collisional boundary between North America and a ribbon continent. Geologic arguments for the existence of the collision boundary in the upper crust are combined with geophysical observations for its deeper structure.

Originality and significance

The differentiation between collisional and accretionary boundaries in ancient

orogens is a question of first order importance. The accretionary and/or non-collisional models are the most widely accepted viewpoints of orogenesis of the North American Cordillera, and a provocative model of Cretaceous collision of a ribbon continent has been published and advocated for by few researchers. Thus, geophysical evidence that can bear on the nature of Cordilleran orogenesis is important. The authors have well described a fairly sharp western boundary in the North American craton, and a significant change in thickness across this boundary. They then interpret the steeply west dipping boundary as a relict Cretaceous collisional boundary.

The authors now address three alternative hypotheses for generating an abrupt step in the lithospheric thickness across what they interpret as a suture. This, in my opinion, is a more balanced approach to the interpretative process and adds to the credibility of their favored interpretation. Please note that on the newly added figure showing the three alternative hypotheses, there are several errors in spelling (potential, delaminated).

Once again, we thank Reviewer #1 for reviewing our paper and acknowledging the scientific contribution of our study. We apologize for the typos in Figure 5 and have thoroughly corrected and improved the figure.

Geodynamic Models

As the crust of the Rocky Mountain fold-thrust belt underwent a minimum of 220 km of shortening, mass balance considerations require a commensurate amount of shortening of the mantle lithosphere. And yet, the mantle lithosphere under the Canadian Cordillera is quite thin. (An alternative is that shortening in the fold-thrust belt is balanced by accretion of terranes at the plate margin following the concept of orogenic float. Orogenic float does not provide a viable mechanism for mass balance in the US Cordillera, so I will, for the sake of argument, discount it for the Canadian Cordillera).

This then requires a mechanism for loss of mantle lithosphere that was thickened to achieve the now relatively thin state, with ablative subduction, vigorous thermal or viscous thinning, or delamination as possibilities. Thus, what may be imaged is the geometry of the destructive feature, not a collisional boundary.

The authors provided the following response to the comment above: "We thank Reviewer #1 for pointing out the potential mass balance problem associated with shortening of the Cordilleran foreland fold-and-thrust belts. However, we point out that a significant mantle thickening is not required to accommodate 220 km of shortening in the thin-skinned (2 to 4 km stratigraphic succession) fold-and-thrust belt. A simple geometric calculation shows only 10 km of distributed shortening within a 70 km thick mantle section balances observed shortening in the fold-and-thrust belt. On the scale of the Cordilleran orogen, such a small amount of shortening would not result in any significant mantle thickening".

I find this logic to be flawed. For strain compatibility, and mass balance, each layer in the lithosphere has to experience the same amount of strain. If the upper crust experiences 220 km of shortening, then any arbitrary layer within the mantle lithosphere will also need to experience 220 km of shortening to maintain strain compatibility. The attempt at area balance in the response is incorrect.

We thank Reviewer #1 for raising this question. We would like to clarify our earlier argument on the balancing problem using a hypothetical model. A thrust fault characterized by a 10 km offset of a layer at a shallow crustal layer can be balanced at a deeper structural layer by a 10 km fault duplex in which no individual fault has more than 1 km of offset. Collectively these faults balance the 10 km of offset at a shallower structural level, but without any layer being offset more than one km. An area balance of the duplex by cross-sectional area palinspastically balances the 10 km offset observed at a shallower crustal level. Extending to a lithospheric scale, the crustal faults can be balanced by distributed shortening within the mantle. As a result, we suggest that the full amount of shortening as

recorded in the fold-and-thrust belt (220 km) may not be required by each layer in the mantle lithosphere.

Regarding issues of the preservation of a lithospheric step at the craton edge, arguments are put forward for the requirement of different mantle lithospheres on either side of the boundary to both generate and preserve this prominent step. This is actually a critical point and is perhaps the most important interpretation in differentiating between the 3 different hypotheses. Differences in rheology, due to differences in chemical depletion and water content are discussed within the framework of collision of two different lithospheres. However, is it not possible to generate differences in rheologic (and geophysical) response simply due to the differences in geologic history and lithospheric modification between the Cordillera and the craton? The Cordillera experienced a geologic history of rifting followed by a prolonged location in a suprasubduction zone environment. The lithosphere at the craton margin experienced rifting and attenuation during the breakup of Rodinia – with all the associated processes of lithospheric modification -- and then protracted modification by processes associated with subduction including interaction with volatiles from dehydrating slabs, convective eddies ... etc. Can the hypothesis that the rheological differences result from these processes superimposed on the same lithosphere be falsified?

We thank Reviewer #1 for pointing out this important argument. The rheological difference plays a major role in the preservation of the sharp CCB as observed in our model. In view of the destructive model, viscous thermal erosion initiated due to craton rifting and associated asthenosphere upwelling, combined with hydration of the mantle by an adjacent subduction zone (Hyndman et al., 2005). This modified the rheology of the craton margin and enhanced the compositional difference across the boundary during prolonged subduction. The rheological difference therefore can result from pure tectonic modification processes without resorting to distinctive continents (as in the collisional model). However, this model cannot readily explain our observations regarding the nature of the lithosphere beneath the Rocky Mountain fold and thrust belt. The new observations imply either: 1) the present-day craton margin (CCB) is a transient feature, which contradicts with the temporal constraints in our study that indicate that its location has not significantly changed over the last 100 Ma; or 2) additional factors are required to stabilize the CCB (such as a long-lived rheological contrast

within the cratonic mantle lithosphere). We added the following discussions on lines 276-292 in the revised manuscript:

“A prediction of the accretionary model is that the (primitive) sub-Cordilleran mantle lithosphere is composed of dry and buoyant rocks that are intrinsically resistive to erosion (Figure 5b). Therefore, strength reduction is needed to promote thinning, possibly through continent rifting followed by refertilization and/or slab dehydration above an east-dipping subduction zone (Hardebol, Pysklywec, & Stephenson, 2012; Hyndman et al., 2005). A direct consequence of the long-term (since Devonian; Figure 5b) lithospheric weakening/removal is the formation of a sharp rheological CCB that acts as a backstop for the migration of the erosion front. However, the foreland location of the CCB is not readily explained by the prolonged destructive process unless one attributes the CCB to a transient feature. The argument of a temporary CCB is difficult to reconcile with 1) a long-lived craton margin supported by the seismic and thermal constraints in this study; and 2) the limited geological evidence of metamorphosed rocks near RMT: phyllite and slate dominate the mountains both east and west of the trench, indicating minimal lithospheric thinning in this region (Cook et al., 1992; Price, 1981). Additional (anti-erosional) factors (e.g., a long-lived rheological contrast within the cratonic mantle lithosphere) are required for the preservation of a relic margin”

Geological Constraints

Application of the hypothesis of a ribbon continent colliding with the North America craton through westward subduction faces challenges in the geologic constraints. As mentioned previously, I am not an expert on the geology of southern Canada but I have the following concerns with the application of the hypothesis south of the Canadian-US border (and I note that section A-A' spans the border). The proposed suture would separate the Windermere Group and its US equivalents from the carbonate passive margin, placing the Windermere within the posited ribbon continent. In the US Cordilleran fold-thrust belt, progressive changes in sedimentary facies and thicknesses can be traced from the craton into the western passive margin between successive thrust sheets, showing a coherent, westward-thickening passive margin sedimentary prism, and leaving no room for juxtaposition of a ribbon continent. Even within single thrust sheets – such as the Willard sheet in Utah and Wyoming – these E to W progressive changes related to the passive margin can be seen, with westward thickening strata and the continuation of “cratonic” Paleozoic strata depositionally overlying the wedge-shaped Neoproterozoic to Early Cambrian

Windermere-affinity sedimentary prism. Two of many examples would be the thrust belt near Las Vegas (see Burchfiel et al., 1974 GSAB), and the Idaho-Utah-Wyoming salient of the thrust belt (see Yonkee and Weil, 2015, ESR).

We thank Reviewer #1 for additional comments on geological constraints. In fact, much of the geological arguments on the correlation and continuation of stratigraphy have been addressed in a series of publications that go back at least to 2001 (Johnston, 2001; Johnston & Borel, 2007; Johnston, 2008; Hildebrand, 2009, 2013, 2014) hence we only briefly summarize the key points here. The correlation of sedimentary strata across the Rocky Mountains has been and continues to be used to argue that there can be no suture separating a ribbon continent from the autochthon. This argument confuses correlation with observation and ignores much data including: 1) there is a shale basinal domain that separates shallow water sedimentary sequences east and west of a carbonate-shale (C-S) facies boundary that characterizes the entire Cordilleran foreland, and hence one cannot walk shallow water sedimentary sequences directly from the fold-and-thrust belt into the hinterland region of the Cordillera; 2) Cambrian and younger shallow water carbonate sequences in the hinterland are characterized by faunal assemblages that are distinct from shallow water carbonates east of the C-S boundary; 3) the basement west to the sedimentary sequences west of the C-S boundary (sampled in Mesozoic diatremes and intrusive pipes) is demonstrably different (younger and probably largely Grenvillian in age) from the basement to the sedimentary succession east of the C-S boundary (Archean to Paleoproterozoic in age).

The second inconsistency is in the timing of emplacement of the thrust sheets with Windemere-related strata. Two thrusts that emplace the Windemere-related rocks include the Wheeler Pass thrust in the Spring Mountains in Nevada and the Willard thrust in Utah. The former is a Late Jurassic structure (Giallorenzo et al., 2018, GSAB) and the later initiated slip at ca. 130 Ma (Eleogram, 2014; Gentry, 2016, and many others), and there is no evidence that they were emplaced in a Late Cretaceous collisional event.

We would like to point out that these thrust sheets imbricate strata that we interpret to be contained within the Cordilleran ribbon continent. Hence these faults are considered as the pre-existing structures that were formed prior to the terminal collision of ribbon continent with the North American craton. In other words, their early emplacement age does not contradict with the collisional model. As an example, the cross-section of Willard thrust shows that the Precambrian and Paleoproterozoic strata in the hanging wall of the fault are not present in the footwall (Yonkee & Weil, 2015). The surface trace of the fault marks a significant change in the nature of the Paleozoic passive margin sequence. This transition is

likely the equivalent of what has been mapped as the Carbonate to Shale facies boundary in Canada. Hence, we suggest that the thrust faults lie close to but west of what is possibly the southern continuation of the cryptic surface trace of the suture and are confined within the Cordillera ribbon continent.

With regard to the Belt-Purcell Group, the Belt sits positionally on top of Proterozoic crystalline basement in SW Montana (LaHood facies), and also occurs east of the thrust belt in the Little Belt Mountains; yes, parts of the Belt Supergroup experienced major translation during Mesozoic to early Tertiary thrusting, but the entire basin is not allochthonous. The point here is that it is very difficult to apply the ribbon continent collision model as advocated by Hildebrand to the US Cordillera.

In the last round of revision, we pointed out that the Belt-Purcell Supergroup is everywhere in the hangingwall of the major thrust fault and does not occur in the footwall of the Purcell Thrust (Sears, 2007), which supports the allochthonous nature of the Belt-Purcell Supergroup and restores it to west of the C-S boundary. As far as the Little Belt Mountains succession, it lies east of the C-S boundary and is characterized by a stratigraphy that is distinct from the Belt-Purcell stratigraphy. The comparison of strata columns from the two regions (Obradovich & Peterman, 1968) show that the amount of overlap between the two sequences is limited (much of the of strata contained in the Belt-Purcell sequence is not present in Little Belt mountains). In addition, the Little Belt sequence lies unconformably over crystalline basement whereas the Belt-Purcell terminates downward against the Lewis thrust. Correlation of the two sequences is permissive, but an equally valid interpretation is that the differences between the two successions imply that they were not contiguous during deposition. There are also aspects of the Belt-Purcell Supergroup that are difficult to reconcile with its interpretation as being North American including presence of detrital zircons that have no North American source; and the local presence of basement rocks within the Purcell Thrust sheet that have no known correlatives in the Laurentian basement (Canadian Shield).

Overall, we acknowledge that the geological constraints are of great importance to validate/falsify the tectonic model. However, as have been shown in recent publication records and this revision, there has been much debate on the level of acceptance in terms of geological data. For example, it has been shown in Johnston (2008) that there are abundant geological, paleontological and paleomagnetic data that are difficult to reconcile within the existing paradigm. The broader Cordilleran community has, however, not accepted the new evidence and data. As an example, although the paleomagnetic data demonstrating that much of the Cordillera is far-travelled with respect to cratonic North America is abundant and repeatable, the community has simply rejected these data (also pointed out by Reviewer

#2). Hence, it is our wish that the updated observations based on seismic data would enable a new look at the problem and prompt positive discussions within the Cordilleran community.

As to your mention of Paul Hoffman's support for this concept, please note that Paul Hoffman, while an outstanding tectonicist, has no track record of working on the Mesozoic Cordillera of the US. I also note that Paul has played a major role in getting Hildebrand's contributions published by the Geological Society of America, so of course he is in support of geophysical arguments that support this model.

We believe that the comments regarding Reviewer #2 are not productive at this stage. Having a track record (or lack thereof) in a specific geographical area should not be taken into consideration in a scientific debate, only the discussion and weighting of the data and observations should be used to validate an argument. Having reviewers with opposite points of view is part of a healthy scientific debate.

Data & methodology: validity of approach, quality of data, quality of presentation

I am not a geophysicist and so I cannot evaluate these issues. However, following my earlier comment regarding better addressing why this data is unique, different, an improvement, etc, over the prior geophysical data sets that have imaged across this boundary, I think the authors have well addressed these issues.

We thank Reviewer #1 for the comments on the data and method. The uniqueness and significance of the data are much clearer to the reader. We would like to reiterate that we have put significant effort into validating our seismic observations (including the tests shown in the supplementary material). It is precisely these new and critical observations (which differ from earlier lower-resolution seismological approaches) that motivate us, and hopefully the scientific community, to revisit the existing tectonic theories and assumptions pertaining to the Cordillera-craton transition.

Conclusions: robustness, validity, reliability

The authors have delineated a clear boundary between thicker, seismically fast lithosphere, and thinner, low-velocity lithosphere to the west. Although I am not a geophysicist, the methods, documentation, and resolution of this feature seems very robust. I am less convinced of the interpretation of this boundary. The boundary is slightly west dipping in profile A-A', but is steep to east-dipping in profile B-B', and yet the authors embrace the west-dipping geometry. The geologic predictions as applied to the US Cordillera do not match the geologic record, despite the numerous publications by Hildebrand. If the advocated model would still be valid if it did not make predictions south of the international border and be specifically relevant to the Canadian Cordillera, then I do not have the expertise to assess this validity and would recommend a review be solicited from someone like Ray Price. In either case, the authors do provide geophysical evidence that permits a collisional hypothesis, and although I am not convinced of the validity of this hypothesis to the western Cordillera, it is provocative and will provide a platform for future work in testing the ribbon continent collision model

We thank Reviewer #1 for confirming the contribution of our study. First of all, we agree that the geology of the Cordillera is demonstrably continuous across the 49th parallel and emphasize that data collected from both sides of the border should be equally weighted in models and interpretations regarding the tectonic evolution and large-scale structure of the orogen. As pointed out by reviewer, there is a significant change in dip from AA' to BB' profiles. The transition in boundary dip agrees well with the change in fault motion along the RMT. In collisional model, this is interpreted as a result of changing the direction of convergence between the Cordillera ribbon continent and North America. We have clarified this point on lines 231-237 in the revised manuscript as below:

“The transition from convergent to strike-slip motion coincides with the change of dipping direction (i.e., westward to sub-vertical/eastward; Figure 4), implying a dominant margin-parallel component of transpressive motion of the Cordillera relative to the craton in this region. This argument is corroborated by an overlapped surface suture and mantle boundary (Figure 3a) and supports the interpretation of the TF as a lithosphere penetrating structure⁴⁹.”

Second, Cordilleran orogenesis is a controversial research topic and has been subject of active debate for decades. As stated earlier we intend to prompt this debate by presenting the robust geophysical data, the core of this study; we agree that the interpretation of CCB is

subject to discussion. As pointed out by Reviewer #1, there is a lack of consensus upon interpretations based on observations made in the US, as exemplified by the publications of Hildebrand who proposed a ribbon continent interpretation of the Cordillera based on interpretation of observations made in the Cordillera of the conterminous US. As such, future investigations from a joint effort of geoscience community is required to fully address these controversies, which cannot be addressed for the moment and is beyond the scope of our paper. To facilitate future research, we have proposed a critical factor for testing the collisional model in the revised manuscript, which could be found on lines 302-304 as below:

“The collisional process predicts a crustal suture in the foreland, the identification of which will be critical for substantiating this hypothesis and requires high-resolution seismic imaging of the crustal structures.”

We thank Reviewer #1 again for taking time to review our paper.

Reviewer #2 (Remarks to the Author):

The authors have satisfactorily answered the comments of all reviewers and their paper, which provides new seismological findings of importance for any interpretation of Cordilleran tectonic development in western North America, should be accepted without further revision for publication in Nature Communications.

I have discussed the lithostratigraphic correlations said to tie the outboard terranes with cratonic Laurentia with the geologists responsible. Their interpretations may be correct but alternative interpretations are also possible: critical areas are heavily forested with poor bedrock exposure. The basic mapping was done decades ago when alternative interpretations were seldom considered.

Cordilleran geologists who two decades ago dismissed without explanation a mountain of paleomagnetic evidence should not now be able to block soundly-based (pun unintended) geophysical data.

The tectonic interpretation offered by the authors may also be non-unique, but

it is the most parsimonious one in light of existing data. The paper is also valuable for exposing the weaknesses and contradictions in alternative interpretations involving cratonward-dipping subduction beneath western Laurentia since the mid-Mesozoic.

The westward-dipping Cretaceous subduction-collision model is not without its own problems (e.g., dynamic subsidence and uplift history of the Western Interior basin). These lie outside the scope of the present paper, which has already been delayed for over half a year since my earlier review.

We thank Reviewer #2 for endorsing our study. There has been a significant progress in understanding Cordilleran tectonics over the past decades. However challenges still exist to fully reconcile geophysical observations with geological constraints. This study intends to promote more discussions in the community and offers new direction to future multi-disciplinary investigations.

Reviewer #3 (Remarks to the Author):

I reviewed a previous version of this manuscript (as Reviewer #3) and believe it has been improved especially in terms of discussing and evaluating previous model (such as Bao et al.) and a more justified interpretation of the tomographic images, but there are still a few issues that need to be addressed.

I did note that Currie is a co-author which is why I was surprised that her prior work was not discussed in more detail. I am happy to see they are now.

We thank Reviewer #3 for constructive comments that improved the discussion and presentation of our paper.

So what is really the difference between the CCB and RMT? As it is written I don't see the main distinction. Can they be the same? Or are they fundamentally different tectonic features?

We thank Reviewer #3 for raising this question. These terms were not clearly differentiated in the earlier manuscript, and we have modified the text to clarify this. The location and formation age of these two structures are distinctively different. First, a critical argument in our paper is that the CCB represents the upper mantle extension of the facies boundary that lies directly to the east of the RMT. The origin of RMT has been debated and has been generally interpreted as an Eocene extensional fault (*Cook et al.*, 1992). We have defined the RMT more clearly on lines 65-70 of the revised manuscript, which read as

“Based on a range of geological and geophysical (primarily paleomagnetic) observations⁴, the suture (boundary) is assumed to run along, or adjacent to, a carbonate-shale (C-S) facies boundary directly east of the Rocky Mountain Trench (RMT), an orogen-parallel valley that extends from Montana to Yukon and which in the south is primarily a product of Cenozoic normal-faulting (Figure 1).”

The age of the RMT (Post-Eocene) is much younger than the CCB (Late Cretaceous), which potentially represents a reactivated structure following the collision. We have clarified this point in our discussions as below (see lines 212-217 of the revised manuscript):

“Following the shortening, the release of compressive stress near the thrust termination in the foreland reactivated the basal décollement, causing post-Eocene normal faulting in the southern RMT⁴³ and regional extension of up to 25 km⁴⁵. The close spatial association of the RMT with the CCB implies that it results from minor reactivation of the CCB during extension.”

In the delamination model (*Bao et al.*, 2014), the location of delamination is suggested to be controlled by the proto step (lithospheric weak zone) underneath the RMT. This argument implies that the RMT is a pre-delamination structure. However, the proposed relationship between RMT and CCB (due to delamination) mistakes the age of these two structures (i.e., delamination occurs before the formation of the RMT), we point out this issue on lines 263-265 of the revised manuscript:

“Although this provides a plausible explanation for the spatial affinity between the CCB and the RMT, the normal faults of the RMT region are too young to be a controlling factor of the suggested delamination...”

Line 84. Delete "unique"

Line 88-89. Delete ", and thus sheds new light on the nature of the southern Canadian Cordillera"

Line 94. Ref 32? Wrong reference?

Line 101. replace "presumed" with "interpreted"

We thank Reviewer #3 for the suggestions on the writing. These have been corrected in the revised manuscript.

Lines 120-122 and Figure 2. The color scale in Figure 2 does not extend to 2.5% (or 3%). The colors are saturated or you are citing the wrong values in one place. And in the text the velocity perturbations should be a negative or more specially say a reduction.

This a good observation by Reviewer #3. The colors of figure are saturated for better illustration of velocity contrast across the CCB. The velocity measurements in the vicinity of the CCB are shown in supplementary Figure 11 (also see below). The velocity beneath the Cordillera could be as low as -2.5% and -3% for P and S models, respectively. In fact, the average velocity could be even lower after correcting for damping effects. We have changed the text (see below and lines 120-123 of the revised manuscript) and use 'negative' and 'positive' to refer to low and high velocity structures.

*"Beneath the southern Canadian Cordillera, **negative** velocities of -2.5% (-3%) relative to the reference model³⁴ for P (S) waves extend to 300 km depth (Figure 2a). To the east, **positive** velocities..."*

Figure R1. Measurements of peak velocities on either side of the transition boundary for (a) P and (b) S models. The black lines show the raw measurements from the model. The red lines show the corrected values after compensating for the amplitude underestimate based on the results of hypothesis tests.

Line 126. What subsurface extension? Inferred from what data or study? This is unclear.

We thank Reviewer #3 for pointing out this issue. We have decided to remove these discussions on the craton structure since they distract readers' attention from the CCB. The structure of the craton and its inference on the tectonic evolution of western Laurentia have been presented in two separate papers (*Chen et al.*, 2017, 2018).

Line 129. Delete "the focus of this study"

Deleted as suggested.

Figure 4. If this is really the tomographic model presented along those slices (which I assume it is) and not just a cartoon, then why not use exactly the same color scale and range as you do in Figure 2? Are the two profiles in the same position as those in Fig2B? You still are using the term "subduction" in the figure which is fundamentally incorrect.

Figure 4 (cont). The northern profile in the 3D figure has a low velocity subhorizontal feature around 200 km depth. this is not present in the cross sections in Figure 2B. What is this? Why haven't you addressed this? I see it in the depth slices in Figure 2A.

Reviewer #3 is right that this figure presents the actual velocities and was made by using open-source software Paraview. The color scheme is the default one in the software. The locations of the northern and southern profiles in Figure 4 are not the same as those presented in Figure 2, due to difficulties in controlling cross-section locations in the software (which requires to input origin and normal direction of the profile instead of the coordinates of two end points as used in Figure 2). In Figure 2, the northern profile did not cross the low velocity structure in central Alberta. This low velocity zone is a pronounced structure residing above 150 km depth, which was interpreted as strongly reworked/refertilized mantle lithosphere (Hearne craton) during the Proterozoic assembly of western Laurentia (*Chen et al.*, 2017). To address Reviewer #3's concerns, we have improved this figure by 1) using the same color scheme and range as those used in Figure 2, 2) extracting the same cross-sections as those in Figure 2, and 3) removing the "subduction" from the plot.

I would remove the Hildenbrand and Whalen reference as it doesn't really help your story. It is wild as I mentioned previously. And on a related point, now that you have toned down the "subduction" interpretation of the craton I don't think the westward subduction references or discussion really adds to your interpretation. The ribbon continent collision idea needs to be better supported in the first half of the paper. Much of the text still remains from the previous version that supports the westward subduction idea that you have now more or less discarded.

We thank Reviewer #3 for making this suggestion. We have removed this reference from the revised manuscript. We have also ensured that the text does not use phrases that hint on **the westward subduction of the North America continent**. However, we believe that the westward subduction idea is still a valid point in our paper, as it is an essential element in the collisional model. To avoid confusion, we have clarified, wherever possible, that it is the westward subduction of oceanic plates (not the north American continent) that is required for the collisional model. For example, on lines 43-46 we suggest that

"Alternate hypotheses favor episodes of westward subduction of oceanic plates that produced the Cordilleran composite (upper plate) in the form of intra-oceanic arcs (i.e., Insular terrane)¹⁶ or a preassembled micro-continent^{4,11,17} prior to collision with the craton."

and on lines 62-65 we suggest that

"The boundary potentially preserves an oceanward (i.e., westward) dipping geometry of a relic craton margin following the break-off of a westward subducting oceanic plate^{4,17} (Figure 1b)."

We thank Reviewer #3 again for the constructive comments that enhance the presentation and impact of our paper.

References

- Bao, X., Eaton, D. W., & Guest, B. (2014). Plateau uplift in western Canada caused by lithospheric delamination along a craton edge. *Nature Geoscience*, 7(11), 830–833. <https://doi.org/10.1038/ngeo2270>
- Chen, Y., Gu, Y. J., & Hung, S.-H. (2017). Finite-frequency P-wave tomography of the Western Canada Sedimentary Basin: Implications for the lithospheric evolution in Western Laurentia. *Tectonophysics*, 698, 79–90.
- Chen, Y., Gu, Y. J., & Hung, S. H. (2018). A New Appraisal of Lithospheric Structures of the Cordillera-Craton Boundary Region in Western Canada. *Tectonics*. <https://doi.org/10.1029/2018TC004956>
- Cook, F. A., Varsek, J. L., Clowes, R. M., Kanasewich, E. R., Spencer, C. S., Parrish, R. R., ...

- Price, R. A. (1992). Lithoprobe crustal reflection cross section of the southern Canadian Cordillera, 1, Foreland thrust and fold belt to Fraser River fault. *Tectonics*, *11*(1), 12–35.
- Hardebol, N. J., Pysklywec, R. N., & Stephenson, R. (2012). Small-scale convection at a continental back-arc to craton transition: Application to the southern Canadian Cordillera. *Journal of Geophysical Research: Solid Earth*, *117*(1), 1–18.
<https://doi.org/10.1029/2011JB008431>
- Hildebrand, R. S. (2009). Did westward subduction cause Cretaceous–Tertiary orogeny in the North American Cordillera? *Geological Society of America Special Papers*, *457*, 1–71.
- Hildebrand, R. S. (2013). *Mesozoic assembly of the North American cordillera* (Vol. 495). Geological Society of America.
- Hildebrand, R. S., & Whalen, J. B. (2014). Arc and slab-failure magmatism in Cordilleran Batholiths I—The Cretaceous Coastal Batholith of Peru and its role in South American orogenesis and hemispheric subduction flip. *Geoscience Canada*, *41*(3), 255–282.
- Hyndman, R. D., Currie, C. A., & Mazzotti, S. P. (2005). Subduction zone backarcs, mobile belts, and orogenic heat. *GSA Today*, *15*(2), 4–10.
- Johnston, S. T. (2001). The Great Alaskan Terrane Wreck: reconciliation of paleomagnetic and geological data in the northern Cordillera. *Earth and Planetary Science Letters*, *193*(3), 259–272.
- Johnston, S. T. (2008). The Cordilleran Ribbon Continent of North America. *Annu. Rev. Earth Planet. Sci.*, *36*(January), 495–530. <https://doi.org/10.1146/annurev.earth.36.031207.124331>
- Johnston, S. T., & Borel, G. D. (2007). The odyssey of the Cache Creek terrane, Canadian Cordillera: Implications for accretionary orogens, tectonic setting of Panthalassa, the Pacific superwell, and break-up of Pangea. *Earth and Planetary Science Letters*, *253*(3), 415–428.
- McLellan, M., Schaeffer, A. J., & Audet, P. (2018). Structure and fabric of the crust and uppermost mantle in the northern Canadian Cordillera from Rayleigh-wave tomography. *Tectonophysics*.
- Obradovich, J. D., & Peterman, Z. E. (1968). Geochronology of the Belt series, Montana. *Canadian Journal of Earth Sciences*, *5*(3), 737–747.
- Polat, A., Frei, R., Longstaffe, F. J., Thorkelson, D. J., & Friedman, E. (2018). Petrology and geochemistry of the Tasse mantle xenoliths of the Canadian Cordillera: A record of Archean to Quaternary mantle growth, metasomatism, removal, and melting. *Tectonophysics*, *737*, 1–26.
- Price, R. A. (1981). The Cordilleran foreland thrust and fold belt in the southern Canadian Rocky Mountains. *Geological Society, London, Special Publications*, *9*(1), 427–448.
- Yonkee, W. A., & Weil, A. B. (2015). Tectonic evolution of the Sevier and Laramide belts within the North American Cordillera orogenic system. *Earth-Science Reviews*, *150*, 531–593.
- Zaporozan, T., Frederiksen, A. W., Bryksin, A., & Darbyshire, F. (2018). Surface-wave images of western Canada: lithospheric variations across the Cordillera–craton boundary. *Canadian Journal of Earth Sciences*, (999), 1–10.

REVIEWERS' COMMENTS:

Reviewer #1 (Remarks to the Author):

This is now my third review of this contribution. As I pointed out previously, the manuscript now leans towards more balance by considering alternative mechanisms for development of an abrupt change in lithospheric thickness and in apparent rheological/density properties. However, I don't think it treats alternative hypotheses fairly. In my opinion, despite the authors attempt to convince me otherwise, the Late Cretaceous collisional model of a ribbon continent is not compatible with the geologic observations of the western U.S.. Note that this is not a criticism of the geophysics, but a criticism of the interpretation of the geophysics as supporting the ribbon continent collision model. I don't think this treatment of the ribbon continent collision model helps to move the dialog forward. The paper would be much more tenable if it was left that the refined resolution of the CCB could be interpreted as A, B, or C, rather than going out on a limb to advocate for a model that is not supported by the surface geology. For this reason, I recommend rejection of this paper.

Some specific objections that have been discussed in the reviews and rebuttals include:

1. I previously pointed out that the timing of motion for the major quartzite-dominated or dominant thrusts in the Sevier fold thrust belt, including the Willard, Canyon Range, Wah Wah, and Wheeler Pass thrust do not support the proposed mid- to Late Cretaceous timing for the collision. The Wheeler Pass is a Late Jurassic thrust and the Willard began motion at 130 Ma. The Willard thrust motion and exhumation record is supported by thermochronometry of the thrust sheet coupled with the record of synorogenic deposits in its footwall. I find the authors suggestion that these structures pre-date the suturing process of a ribbon continent as untenable, because the record of the earlier thrust comes from complementary data sets in both the hanging wall and footwall, that is, on either side of the proposed suture.
2. I previously pointed out that the stratigraphy in the thrust belt can be progressively tracked between each successive thrust sheet to reconstruct a westward thickening wedge. Yes, most of the Windermere exposures are from the hanging wall of the dominant thrust sheets, but there are exposures in the footwall as well, including at Antelope Island in Great Salt Lake. Other compelling ties include the presence of 490-500 Ma detrital zircons in the Cambrian Worm Creek Quartzite and its equivalents that are present in the Paris thrust sheet (Willard equivalent) as well as in the foreland from central Wyoming to Montana. These ages and their Hf isotopes match hypabyssal alkalic intrusions in Idaho (Link et al., 2017). Similar arguments can be made using Ordovician strata, as well Neoproterozoic strata (see extensive DZ dataset presented in Matthews et al., 2017 that is used to argue against the ribbon continent hypothesis).

The logic regarding the strain compatibility and structural balance still remains flawed. This is not a major component of the paper, but the authors dismiss my earlier criticism using continued flawed logic. This does not strengthen my opinion of the other interpretations of the authors.

For clarification, I reiterate my comments and the authors response below.

Review comment 1. As the crust of the Rocky Mountain fold-thrust belt underwent a minimum of 220 km of shortening, mass balance considerations require a commensurate amount of shortening of the mantle lithosphere. And yet, the mantle lithosphere under the Canadian Cordillera is quite thin. (An alternative is that shortening in the fold-thrust belt is balanced by accretion of terranes at the plate margin following the concept of orogenic float. Orogenic float does not provide a viable mechanism for mass balance in the US Cordillera, so I will, for the sake of argument, discount it for the Canadian Cordillera). This then requires a mechanism for loss of mantle lithosphere that was thickened to achieve the now relatively thin state, with ablative subduction, vigorous thermal or viscous thinning, or delamination as possibilities. Thus, what may be imaged is the geometry of the destructive feature, not a collisional boundary.

Authors reply 1. "We thank Reviewer #1 for pointing out the potential mass balance problem associated with shortening of the Cordilleran foreland fold-and-thrust belts. However, we point out that a significant mantle thickening is not required to accommodate 220 km of shortening in the thin-skinned (2 to 4 km stratigraphic succession) fold-and-thrust belt. A simple geometric calculation shows only 10 km of distributed shortening within a 70 km thick mantle section balances observed shortening in the fold-and-thrust belt. On the scale of the Cordilleran orogen, such a small amount of shortening would not result in any significant mantle thickening".

Review comment 2: I find this logic to be flawed. For strain compatibility, and mass balance, each layer in the lithosphere has to experience the same amount of strain. If the upper crust experiences 220 km of shortening, then any arbitrary layer within the mantle lithosphere will also need to experience 220 km of shortening to maintain strain compatibility. The attempt at area balance in the response is incorrect.

Authors reply 2. "We thank Reviewer #1 for raising this question. We would like to clarify our earlier argument on the balancing problem using a hypothetical model. A thrust fault characterized by a 10 km offset of a layer at a shallow crustal layer can be balanced at a deeper structural layer by a 10 fault duplex in which no individual fault has more than 1 km of offset. Collectively these faults balance the 10 km of offset at a shallower structural level, but without any layer being offset more than one km. An area balance of the duplex by cross-sectional area palinspastically balances the 10 km offset observed at a shallower crustal level. Extending to a lithospheric scale, the crustal faults can be balanced by distributed shortening within the mantle. As a result, we suggest that the full amount of shortening as recorded in the fold-and-thrust belt (220 km) may not be required by each layer in the mantle lithosphere.

Review comment 3: The logic, unfortunately, remains flawed. Note that, in their duplex example above, the layer with a 10 fault duplex experiences 10 km of shortening by 10 separate faults with 1 km of displacement each. Thus, it is incorrect to state "without any layer being offset more than one km". In a duplex, the same layer gets repeatedly imbricated, with a floor and roof thrust. In this thought experiment, the layer is offset 10 km, not 1 km. Thus, it is incorrect to conclude that "As a result, we suggest that the full amount of shortening as recorded in the fold-and-thrust belt (220 km) may not be required by each layer in the mantle lithosphere". To reiterate the fundamental concept of strain compatibility, to maintain strain compatibility, each layer must be shortened an equivalent amount within the lithospheric column. This is a fundamental tenant in constructing any geodynamic model.

Comments regarding the Belt-Purcell Group

Review comment 2. With regard to the Belt-Purcell Group, the Belt sits depositionally on top of Proterozoic crystalline basement in SW Montana (LaHood facies), and also occurs east of the thrust belt in the Little Belt Mountains; yes, parts of the Belt Supergroup experienced major translation during Mesozoic to early Tertiary thrusting, but the entire basin is not allochthonous. The point here is that it is very difficult to apply the ribbon continent collision model as advocated by Hildebrand to the US Cordillera.

Authors reply 2. In the last round of revision, we pointed out that the Belt-Purcell Supergroup is everywhere in the hangingwall of the major thrust fault and does not occur in the footwall of the Purcell Thrust (Sears, 2007), which supports the allochthonous nature of the Belt-Purcell Supergroup and restores it to west of the C-S boundary. As far as the Little Belt Mountains succession, it lies east of the C-S boundary and is characterized by a stratigraphy that is distinct from the Belt-Purcell stratigraphy. The comparison of strata columns from the two regions (Obradovich & Peterman, 1968) show that the amount of overlap between the two sequences is limited (much of the of strata contained in the Belt-Purcell sequence is not present in Little Belt mountains). In addition, the Little Belt sequence lies unconformably over crystalline basement whereas the Belt-Purcell terminates downward against the Lewis thrust. Correlation of the two sequences is permissive, but an equally valid interpretation is that the differences between the two

successions imply that they were not contiguous during deposition. There are also aspects of the Belt-Purcell Supergroup that are difficult to reconcile with its interpretation as being North American including presence of detrital zircons that have no North American source; and the local presence of basement rocks within the Purcell Thrust sheet that have no known correlatives in the Laurentian basement (Canadian Shield).

Review comment 3. In my last review I pointed out examples of where the Belt occurs in the footwall of the major thrusts that elsewhere carry Belt strata. The authors chose to either not address them (LaHood facies) or to provide ad hoc arguments as to why the example I provided (Little Belt Mountains) should be discounted. I remain unconvinced that the Belt strata in the Little Belt Mountains is not part of the Belt supergroup. My point is that in this eastern locality, it is sitting positionally on crystalline basement. That is not a reason to discount it as Belt. To do so follows a circular argument (e.g., if not allochthonous, cannot be Belt; therefore Belt is allochthonous). Also, sedimentary sequences at the margin of basins, where not fault bounded, tend to be thinner, and thus all members may not be present. This is not a contradiction.

Invoking of reputation of other reviewers

Authors comment 1.

The suggestion that the ribbon continent model is 'not taken seriously' is entirely inconsistent with the record of publications both of ribbon continent interpretations of the Cordillera and of papers citing and testing the ribbon continent model. It also seems reasonable to point out that the other two reviewers appear to take the ribbon continent model seriously, one of whom is Paul Hoffman (Reviewer #2), a world-renowned tectonicist.

Review comment 2. As to your mention of Paul Hoffman's support for this concept, please note that Paul Hoffman, while an outstanding tectonicist, has no track record of working on the Mesozoic Cordillera of the US. I also note that Paul has played a major role in getting Hildebrand's contributions published by the Geological Society of America, so of course he is in support of geophysical arguments that support this model.

Authors comment 2. We believe that the comments regarding Reviewer #2 are not productive at this stage. Having a track record (or lack thereof) in a specific geographical area should not be taken into consideration in a scientific debate, only the discussion and weighting of the data and observations should be used to validate an argument. Having reviewers with opposite points of view is part of a healthy scientific debate.

Review comment 3.

This a bait and switch. You first mentioned the reputation of a reviewer as a justification for your interpretation; I did not. You can ask Paul about his role in advocating for Hildebrand's ideas.

Reviewer #3 (Remarks to the Author):

The authors have addressed all the comments sufficiently, and through doing so they have improved the manuscript substantially based on the series of reviews by the 3 reviewers. I'm happy to support its publication.

Meghan S. Miller

Response to reviewers:

This document presents our point-by-point response to the reviewers' comments. The original reviewer comments are in ***Bold Italics*** and call-outs to other papers are denoted in *Italics*.

REVIEWERS' COMMENTS:

Reviewer #1 (Remarks to the Author):

This is now my third review of this contribution. As I pointed out previously, the manuscript now leans towards more balance by considering alternative mechanisms for development of an abrupt change in lithospheric thickness and in apparent rheological/density properties. However, I don't think it treats alternative hypotheses fairly. In my opinion, despite the authors attempt to convince me otherwise, the Late Cretaceous collisional model of a ribbon continent is not compatible with the geologic observations of the western U.S.. Note that this is not a criticism of the geophysics, but a criticism of the interpretation of the geophysics as supporting the ribbon continent collision model. I don't think this treatment of the ribbon continent collision model helps to move the dialog forward. The paper would be much more tenable if it was left that the refined resolution of the CCB could be interpreted as A, B, or C, rather than going out on a limb to advocate for a model that is not supported by the surface geology. For this reason, I recommend rejection of this paper.

Reply: We thank Reviewer 1 for three rounds of constructive reviews. Although Reviewer 1 holds different opinions on the model interpretation, it is encouraging to see that Reviewer 1 acknowledges the new geophysical data and its positive role in prompting the Cordillera debate. First of all, we would like to clarify that we do not intend to treat the various interpretations unfairly. In fact, the fundamental point of our study is to present the seismic data and provide a new angle to objectively assess the existing tectonic paradigms in light of the new geophysical observations. We apologize if the way we organized the paper or the language we used misinformed Reviewer 1 about the objectiveness of our study. To address this issue, we have made revisions to further improve the overall balance of the paper and soften the language in the following ways:

- 1) We changed the title and abstract for a more objective perspective of our study.
- 2) We clearly separated the discussions of three mechanisms. Each hypothesis is now individually assessed per Reviewer #1's suggestion. We also thoroughly rewrote the

discussion of the destructive mechanisms, in particular the thermal/viscous erosion scenario. We emphasized that the existing seismic and thermal constraints cannot rule out the mechanism involving thermal/viscous erosion, a process playing an important role in (re)shaping the CCB characters (e.g., location, geometry and sharpness).

- 3) We provided a more objective assessment of the collisional model. We avoid using words “favor”, “argue”, and “attribute” that may lead to the subjective interpretation and focus more on presenting the seismic observations.
- 4) Although we hold different views on the interpretation of geologic observations, particularly the nature (allochthonous vs. autochthonous) of the Belt-Purcell group and the associated correlative strata, Reviewer #1’s comments from previous rounds indeed encourage us to carefully examine the compatibility of seismic observations to the surface geology in the western US. This was added in a new paragraph for proper discussion.

Finally, we share the same view as the reviewer that an objective assessment of tectonic model should be based upon the reliable data observations present in the study. We favor the collisional model mainly because we believe that this interpretation provides the simplest explanation for the bulk of the available data. We also believe that new constraints and the added perspectives and scrutiny can only increase the global awareness of the uncertainties and challenges, as evidenced by three round of peer reviews, in continuing to uncover the mysteries of Cordillera-Craton formation and evolution.

Some specific objections that have been discussed in the reviews and rebuttals include:

I previously pointed out that the timing of motion for the major quartzite-dominated or dominant thrusts in the Sevier fold thrust belt, including the Willard, Canyon Range, Wah Wah, and Wheeler Pass thrust do not support the proposed mid- to Late Cretaceous timing for the collision. The Wheeler Pass is a Late Jurassic thrust and the Willard began motion at 130 Ma. The Willard thrust motion and exhumation record is supported by thermochronometry of the thrust sheet coupled with the record of synorogenic deposits in its footwall. I find the authors suggestion that these structures pre-date the suturing process of a ribbon continent as untenable, because the record of the earlier thrust comes from complementary data sets in both the hanging wall and footwall, that is, on either side of the proposed suture.

Reply: We thank Reviewer 1 for raising the concern of emplacement timing again. However, Reviewer 1 seems to make the assumption that the thrust faults are the suture and hence that the hangingwall strata are exotic and the footwall strata are North American. We want to clarify that this is not what has been suggested and it is not what we are suggesting. Specifically, if the thrust faults were the suture, then we would not be referring to the suture as cryptic. In fact, the thrust faults record shortening within the ribbon continent in the Jurassic. Our prediction is that somewhere in the footwall of the imbricate thrust stack (not the footwall of the individual thrusts) there is a cryptic, structurally (and stratigraphically?) masked cryptic suture.

2. I previously pointed out that the stratigraphy in the thrust belt can be progressively tracked between each successive thrust sheet to reconstruct a westward thickening wedge. Yes, most of the Windermere exposures are from the hanging wall of the dominant thrust sheets, but there are exposures in the footwall as well, including at Antelope Island in Great Salt Lake. Other compelling ties include the presence of 490-500 Ma detrital zircons in the Cambrian Worm Creek Quartzite and its equivalents that are present in the Paris thrust sheet (Willard equivalent) as well as in the foreland from central Wyoming to Montana. These ages and their Hf isotopes match hypabyssal alkalic intrusions in Idaho (Link et al., 2017). Similar arguments can be made using Ordovician strata, as well Neoproterozoic strata (see extensive DZ dataset presented in Matthews et al., 2017 that is used to argue against the ribbon continent hypothesis).

Reply: We thank Reviewer 1 for pointing these geological constraints. Much (all?) of these records lie within the proposed ribbon continent. We would like to point out that detrital zircons do not provide a definitive test of the ribbon continent model. They do tell us that there is continental crust (the source of the zircons) in both cratonic North America and in the Ribbon Continent, consistent with the previously published models.

As far as Reviewer 1's mentioning of the paper by *Matthews et al. (2017)*, we find that the main conclusion of their study is that it supports the allochthonous nature of the Cordilleran continent. For example, the paper states that:

“Few models for the Mesozoic-Cenozoic evolution of the North American Cordillera fully integrate paleomagnetic and paleobotanical displacement estimates [e.g., Saleeby, 2003; Dickinson, 2004; Evanchick et al., 2007; Gehrels et al., 2009; Jacobson et al., 2011; Yokelson et al., 2015], and those that do have not found broad acceptance [e.g., Chamberlain and Lambert, 1985; Irving et al., 1996; Johnston, 2008; Hildebrand, 2009; Sigloch and Mihalynuk, 2013]. Although the data presented here are not in itself conclusive, our detrital zircon results are

consistent with large northward translations of rocks of the western Canadian Cordillera. As such, we prefer tectonic models that accommodate the paleomagnetic data. Such models should be more thoroughly considered, as they would have important implications for the interpreted history of western North America.”

The logic regarding the strain compatibility and structural balance still remains flawed. This is not a major component of the paper, but the authors dismiss my earlier criticism using continued flawed logic. This does not strengthen my opinion of the other interpretations of the authors.

For clarification, I reiterate my comments and the authors response below.

Review comment 1. As the crust of the Rocky Mountain fold-thrust belt underwent a minimum of 220 km of shortening, mass balance considerations require a commensurate amount of shortening of the mantle lithosphere. And yet, the mantle lithosphere under the Canadian Cordillera is quite thin. (An alternative is that shortening in the fold-thrust belt is balanced by accretion of terranes at the plate margin following the concept of orogenic float. Orogenic float does not provide a viable mechanism for mass balance in the US Cordillera, so I will, for the sake of argument, discount it for the Canadian Cordillera). This then requires a mechanism for loss of mantle lithosphere that was thickened to achieve the now relatively thin state, with ablative subduction, vigorous thermal or viscous thinning, or delamination as possibilities. Thus, what may be imaged is the geometry of the destructive feature, not a collisional boundary.

Authors reply 1. “We thank Reviewer #1 for pointing out the potential mass balance problem associated with shortening of the Cordilleran foreland fold-and-thrust belts. However, we point out that a significant mantle thickening is not required to accommodate 220 km of shortening in the thin-skinned (2

to 4 km stratigraphic succession) fold-and-thrust belt. A simple geometric calculation shows only 10 km of distributed shortening within a 70 km thick mantle section balances observed shortening in the fold-and-thrust belt. On the scale of the Cordilleran orogen, such a small amount of shortening would not result in any significant mantle thickening”.

Review comment 2: I find this logic to be flawed. For strain compatibility, and mass balance, each layer in the lithosphere has to experience the same amount of strain. If the upper crust experiences 220 km of shortening, then any arbitrary layer within the mantle lithosphere will also need to experience 220 km of shortening to maintain strain compatibility. The attempt at area balance in the response is incorrect.

Authors reply 2. “We thank Reviewer #1 for raising this question. We would like to clarify our earlier argument on the balancing problem using a hypothetical model. A thrust fault characterized by a 10 km offset of a layer at a shallow crustal layer can be balanced at a deeper structural layer by a 10 fault duplex in which no individual fault has more than 1 km of offset. Collectively these faults balance the 10 km of offset at a shallower structural level, but without any layer being offset more than one km. An area balance of the duplex by cross-sectional area palinspastically balances the 10 km offset observed at a shallower crustal level. Extending to a lithospheric scale, the crustal faults can be balanced by distributed shortening within the mantle. As a result, we suggest that the full amount of shortening as recorded in the fold-and-thrust belt (220 km) may not be required by each layer in the mantle lithosphere.

Review comment 3: The logic, unfortunately, remains flawed. Note that, in their duplex example above, the layer with a 10 fault duplex experiences 10 km of shortening by 10 separate faults with 1 km of displacement each. Thus,

it is incorrect to state "without any layer being offset more than one km". In a duplex, the same layer gets repeatedly imbricated, with a floor and roof thrust. In this thought experiment, the layer is offset 10 km, not 1 km. Thus, it is incorrect to conclude that "As a result, we suggest that the full amount of shortening as recorded in the fold-and-thrust belt (220 km) may not be required by each layer in the mantle lithosphere". To reiterate the fundamental concept of strain compatibility, to maintain strain compatibility, each layer must be shortened an equivalent amount within the lithospheric column. This is a fundamental tenant in constructing any geodynamic model.

Reply: There might be some miscommunication in this debate. Reviewer 1 is as genuinely puzzled by our response as we are by his/her criticism. We could probably figure this out very quickly were we to sit down together. In fact, a key part of this debate during three rounds of review is whether lithosphere thinning occurred beneath the Canadian Cordillera. We provided geophysical evidence from both previous work and the current study to show that the lithosphere directly beneath the foreland region was not thinned. Therefore, even if the entire lithosphere column thickens due to shortening under the mass balance constraints, the subsequent thinning process either did not occur or did not affect the foreland region.

Comments regarding the Belt-Purcell Group

Review comment 2. With regard to the Belt-Purcell Group, the Belt sits positionally on top of Proterozoic crystalline basement in SW Montana (LaHood facies), and also occurs east of the thrust belt in the Little Belt Mountains; yes, parts of the Belt Supergroup experienced major translation during Mesozoic to early Tertiary thrusting, but the entire basin is not allochthonous. The point here is that it is very difficult to apply the ribbon continent collision model as advocated by Hildebrand to the US Cordillera.

Authors reply 2. In the last round of revision, we pointed out that the Belt-Purcell Supergroup is everywhere in the hangingwall of the major thrust fault and does not occur in the footwall of the Purcell Thrust (Sears, 2007), which supports the allochthonous nature of the Belt-Prucell Supergroup and restores it to west of the C-S boundary. As far as the Little Belt Mountains

succession, it lies east of the C-S boundary and is characterized by a stratigraphy that is distinct from the Belt-Purcell stratigraphy. The comparison of strata columns from the two regions (Obradovich & Peterman, 1968) show that the amount of overlap between the two sequences is limited (much of the of strata contained in the Belt-Purcell sequence is not present in Little Belt mountains). In addition, the Little Belt sequence lies uncomformably over crystalline basement whereas the Belt-Purcell terminates downward against the Lewis thrust. Correlation of the two sequences is permissive, but an equally valid interpretation is that the differences between the two successions imply that they were not contiguous during deposition. There are also aspects of the Belt-Purcell Supergroup that are difficult to reconcile with its interpretation as being North American including presence of detrital zircons that have no North American source; and the local presence of basement rocks within the Purcell Thrust sheet that have no known correlatives in the Laurentian basement (Canadian Shield).

Review comment 3. In my last review I pointed out examples of where the Belt occurs in the footwall of the major thrusts that elsewhere carry Belt strata. The authors chose to either not address them (LaHood facies) or to provide ad hoc arguments as to why the example I provided (Little Belt Mountains) should be discounted. I remain unconvinced that the Belt strata in the Little Belt Mountains is not part of the Belt supergroup. My point is that in this eastern locality, it is sitting depositionally on crystalline basement. That is not a reason to discount it as Belt. To do so follows a circular argument (e.g., if not allochthonous, cannot be Belt; therefore Belt is allochthonous). Also, sedimentary sequences at the margin of basins, where not fault bounded, tend to be thinner, and thus all members may not be present. This is not a contradiction.

Reply: We don't disagree in principle with any of Reviewer 1's comments regarding the Belt-Purcell sequence. We accept the interpretation of the Lahood Facies (apologies that we failed to respond directly to this point last time) of the Belt-Purcell supergroup as being deposited on the adjacent basement rocks. However, these rocks can be interpreted as being in the hangingwall of the Lewis thrust sheet. As pointed out by one of us (*Johnston, 2008*), those basement rocks yield anomalous ages and cannot be correlated with any known Precambrian basement within the autochthon. Interpretation of the Little Belt sequence as being correlative with the Belt-Purcell is reasonable. Reviewer 1 provides acceptable justifications for the interpretation given that the Little Belt sequence is significantly thinner than the Belt-Purcell and its stratigraphy distinguishable. This is, however, our point - correlation of the Belt-Purcell and the Little Belt sequence is an interpretation, and though an acceptable interpretation, in our view this is a flawed interpretation. The previous 'collisional' models, although not explicitly painted in terms of a ribbon continent, were published in the eighties (*Chamberlin & Lambert*) and seventies (*Moore's*) prior to Hildebrand's works.

Invoking of reputation of other reviewers

Authors comment 1. The suggestion that the ribbon continent model is 'not taken seriously' is entirely inconsistent with the record of publications both of ribbon continent interpretations of the Cordillera and of papers citing and testing the ribbon continent model. It also seems reasonable to point out that the other two reviewers appear to take the ribbon continent model seriously, one of whom is Paul Hoffman (Reviewer #2), a world-renowned tectonicist.

Review comment 2. As to your mention of Paul Hoffman's support for this concept, please note that Paul Hoffman, while an outstanding tectonicist, has no track record of working on the Mesozoic Cordillera of the US. I also note that Paul has played a major role in getting Hildebrand's contributions published by the Geological Society of America, so of course he is in support of geophysical arguments that support this model.

Authors comment 2. We believe that the comments regarding Reviewer #2 are not productive at this stage. Having a track record (or lack thereof) in a specific geographical area should not be taken into consideration in a

scientific debate, only the discussion and weighting of the data and observations should be used to validate an argument. Having reviewers with opposite points of view is part of a healthy scientific debate.

Review comment 3.

This a bait and switch. You first mentioned the reputation of a reviewer as a justification for your interpretation; I did not. You can ask Paul about his role in advocating for Hildebrand's ideas.

Reply: We apologize for the citing Reviewer 2 as part of the debate, which is unscientific and unprofessional, and we would like to clarify our point here. We do not intend to use nor have we used any people's advocacy/reputation as a justification for our interpretation. Our interpretation is based on the new geophysical data and the resulting spatio-temporal constraints on the Cordillera-Craton boundary. We just invoked Reviewer 2 as an example in response to Reviewer 1's earlier comment that "Cited references where the ribbon continent model have been applied have cascading inaccuracies to the extent that no US Cordilleran tectonicists take the application of this to the US Cordillera seriously." In our reply we mainly presented a detailed record of scientific literature supporting the collisional hypothesis and Reviewer #2 is only briefly mentioned as a relevant example of a scientist who has acknowledged the viability of alternative interpretations of Cordilleran geology.

We thank Reviewer 1 again for taking time to review our paper. Despite the debate on the interpretation of the geophysical data, we believe that the critical comments by Reviewer 1 have resulted in a more balanced discussion and this has significantly improved the quality of this study. We sincerely hope this study will encourage continued discussions in the scientific community, as the ultimate goal is to gain a better understanding of the Cordillera.

Reviewer #3 (Remarks to the Author):

The authors have addressed all the comments sufficiently, and through doing so they have improved the manuscript substantially based on the series of reviews by the 3 reviewers. I'm happy to support its publication.

Meghan S. Miller

We thank Reviewer 3 for the constructive comments during three rounds of reviews, which significantly improved the impact of this study.